# Well-tuned Simple Nets Excel on Tabular Datasets

**Arlind Kadra**
Department of Computer Science
University of Freiburg
kadraa@cs.uni-freiburg.de

**Marius Lindauer**
Institute for Information Processing
Leibniz University Hannover
lindauer@tnt.uni-hannover.de

**Frank Hutter**
Department of Computer Science
University of Freiburg &
Bosch Center for Artificial Intelligence
fh@cs.uni-freiburg.de

**Josif Grabocka**
Department of Computer Science
University of Freiburg
grabocka@informatik.uni-freiburg.de

## Abstract

Tabular datasets are the last "unconquered castle" for deep learning, with traditional ML methods like Gradient-Boosted Decision Trees still performing strongly even against recent specialized neural architectures. In this paper, we hypothesize that the key to boosting the performance of neural networks lies in rethinking the joint and simultaneous application of a large set of modern regularization techniques. As a result, we propose regularizing plain Multilayer Perceptron (MLP) networks by searching for the optimal combination/cocktail of 13 regularization techniques for each dataset using a joint optimization over the decision on which regularizers to apply and their subsidiary hyperparameters.

We empirically assess the impact of these *regularization cocktails* for MLPs in a large-scale empirical study comprising 40 tabular datasets and demonstrate that (i) well-regularized plain MLPs significantly outperform recent state-of-the-art specialized neural network architectures, and (ii) they even outperform strong traditional ML methods, such as XGBoost.

## 1 Introduction

In contrast to the mainstream in deep learning (DL), in this paper, we focus on tabular data, a domain that we feel is understudied in DL. Nevertheless, it is of great relevance for many practical applications, such as climate science, medicine, manufacturing, finance, recommender systems, etc. During the last decade, traditional machine learning methods, such as Gradient-Boosted Decision Trees (GBDT) [5], dominated tabular data applications due to their superior performance, and the success story DL has had for raw data (e.g., images, speech, and text) stopped short of tabular data.

Even in recent years, the existing literature still gives mixed messages on the state-of-the-art status of deep learning for tabular data. While some recent neural network methods [1, 46] claim to outperform GBDT, others confirm that GBDT are still the most accurate method on tabular data [48, 26]. The extensive experiments on 40 datasets we report indeed confirm that recent neural networks [1, 46, 11] do not outperform GBDT when the hyperparameters of all methods are thoroughly tuned.

We hypothesize that the key to improving the performance of neural networks on tabular data lies in exploiting the recent DL advances on regularization techniques (reviewed in Section 3), such as data augmentation, decoupled weight decay, residual blocks and model averaging (e.g., dropout or snapshot ensembles), or on learning dynamics (e.g., look-ahead optimizer or stochastic weight averaging).

35th Conference on Neural Information Processing Systems (NeurIPS 2021).

Indeed, we find that even plain Multilayer Perceptrons (MLPs) achieve state-of-the-art results when regularized by multiple modern regularization techniques applied jointly and simultaneously.

Applying multiple regularizers jointly is already a common standard for practitioners, who routinely mix regularization techniques (e.g. Dropout with early stopping and weight decay). However, the deeper question of "Which subset of regularizers gives the largest generalization performance on a particular dataset among dozens of available methods?" remains unanswered, as practitioners currently combine regularizers via inefficient trial-and-error procedures. In this paper, we provide a simple, yet principled answer to that question, by posing the selection of the optimal subset of regularization techniques and their inherent hyperparameters, as a joint search for the best combination of MLP regularizers for each dataset among a pool of 13 modern regularization techniques and their subsidiary hyperparameters (Section 4).

From an empirical perspective, this paper is the first to provide compelling evidence that well-regularized neural networks (even simple MLPs!) indeed surpass the current state-of-the-art models in tabular datasets, including recent neural network architectures and GBDT (Section 6). In fact, the performance improvements are quite pronounced and highly significant.[1] We believe this finding to potentially have far-reaching implications, and to open up a garden of delights of new applications on tabular datasets for DL.

Our contributions are as follows:

1. We demonstrate that modern DL regularizers (developed for DL applications on raw data, such as images, speech, or text) also substantially improve the performance of deep multi-layer perceptrons on tabular data.
2. We propose a simple, yet principled, paradigm for selecting the optimal subset of regularization techniques and their subsidiary hyperparameters (so-called *regularization cocktails*).
3. We demonstrate that these regularization cocktails enable even simple MLPs to outperform both recent neural network architectures, as well as traditional strong ML methods, such as GBDT, on tabular data. Specifically, we are the first to show neural networks to significantly (and substantially) outperform XGBoost in a fair, large-scale experimental study.

## 2 Related Work on Deep Learning for Tabular Data

Recently, various neural architectures have been proposed for improving the performance of neural networks on tabular data. TabNet [1] introduced a sequential attention mechanism for capturing salient features. Neural oblivious decision ensembles (NODE [46]) blend the concept of hierarchical decisions into neural networks. Self-normalizing neural networks [29] have neuron activations that converge to zero mean and unit variance, which in turn, induces strong regularization and allows for high-level representations. Regularization learning networks train a regularization strength on every neural weight by posing the problem as a large-scale hyperparameter tuning scheme [48]. The recent NET-DNF technique introduces a novel inductive bias in the neural structure corresponding to logical Boolean formulas in disjunctive normal forms [26]. An approach that is often mistaken as deep learning for tabular data is AutoGluon Tabular [11], which builds ensembles of basic neural networks together with other traditional ML techniques, with its key contribution being a strong stacking approach. We emphasize that some of these publications claim to outperform Gradient Boosted Decision Trees (GDBT) [1, 46], while other papers explicitly stress that the neural networks tested do not outperform GBDT on tabular datasets [48, 26]. In contrast, we do not propose a new kind of neural architecture, but a novel paradigm for learning a combination of regularization methods.

## 3 An Overview of Regularization Methods for Deep Learning

**Weight decay:** The most classical approaches of regularization focused on minimizing the norms of the parameter values, e.g., either the L1 [51], the L2 [52], or a combination of L1 and L2 known as the Elastic Net [63]. A recent work fixes the malpractice of adding the decay penalty term before

---

[1]In that sense, this paper adds to the growing body of literature on demonstrating the sensitivity of modern ML methods to hyperparameter settings [19], demonstrating that proper hyperparameter tuning can yield substantial improvements of modern ML methods [6], and demonstrating that even simple architectures can obtain state-of-the-art performance with proper hyperparameter settings [38].

momentum-based adaptive learning rate steps (e.g., in common implementations of Adam [27]), by decoupling the regularization from the loss and applying it after the learning rate computation [36].

**Data Augmentation:** Among the augmentation regularizers, Cut-Out [10] proposes to mask a subset of input features (e.g., pixel patches for images) for ensuring that the predictions remain invariant to distortions in the input space. Along similar lines, Mix-Up [60] generates new instances as a linear span of pairs of training examples, while Cut-Mix [58] suggests super-positions of instance pairs with mutually-exclusive pixel masks. A recent technique, called Aug-Mix [20], generates instances by sampling chains of augmentation operations. On the other hand, the direction of reinforcement learning (RL) for augmentation policies was elaborated by Auto-Augment [7], followed by a technique that speeds up the training of the RL policy [34]. Recently, these complex and expensive methods were superseded by simple and cheap methods that yield similar performance (RandAugment [8]) or even improve on it (TrivialAugment [41]). Last but not least, adversarial attack strategies (e.g., FGSM [17]) generate synthetic examples with minimal perturbations, which are employed in training robust models [37].

**Ensemble methods:** Ensembled machine learning models have been shown to reduce variance and act as regularizers [45]. A popular ensemble neural network with shared weights among its base models is Dropout [49], which was extended to a variational version with a Gaussian posterior of the model parameters [28]. As a follow-up, Mix-Out [32] extends Dropout by statistically fusing the parameters of two base models. Furthermore, so-called *snapshot ensembles* [21] can be created using models from intermediate convergence points of stochastic gradient descent with restarts [35]. In addition to these efficient ensembling approaches, ensembling independent classifiers trained in separate training runs can yield strong performance (especially for uncertainty quantification), be it based on independent training runs only differing in random seeds (*deep ensembles* [31]), training runs differing in hyperparameter settings (*hyperdeep ensembles*, [55]), or training runs with different neural architectures (*neural ensemble search* [59]).

**Structural and Linearization:** In terms of structural regularization, ResNet adds skip connections across layers [18], while the Inception model computes latent representations by aggregating diverse convolutional filter sizes [50]. A recent trend adds a dosage of *linearization* to deep models, where skip connections transfer embeddings from previous less non-linear layers [18, 22]. Along similar lines, the Shake-Shake regularization deploys skip connections in parallel convolutional blocks and aggregates the parallel representations through affine combinations [15], while Shake-Drop extends this mechanism to a larger number of CNN architectures [56].

**Implicit:** The last family of regularizers broadly encapsulates methods that do not directly propose novel regularization techniques but have an *implicit* regularization effect as a virtue of their 'modus operandi' [2]. The simplest such implicit regularization is Early Stopping [57], which limits overfitting by tracking validation performance over time and stopping training when validation performance no longer improves. Another implicit regularization method is Batch Normalization, which improves generalization by reducing internal covariate shift [24]. The scaled exponential linear units (SELU) represent an alternative to batch-normalization through self-normalizing activation functions [30]. On the other hand, stabilizing the convergence of the training routine is another implicit regularization, for instance by introducing learning rate scheduling schemes [35]. The recent strategy of stochastic weight averaging relies on averaging parameter values from the local optima encountered along the sequence of optimization steps [25], while another approach conducts updates in the direction of a few 'lookahead' steps [61].

## 4 Regularization Cocktails for Multilayer Perceptrons

### 4.1 Problem Definition

A training set is composed of features $\mathbf{X}^{\text{(Train)}}$ and targets $\mathbf{y}^{\text{(Train)}}$, while the test dataset is denoted by $\mathbf{X}^{\text{(Test)}}$, $\mathbf{y}^{\text{(Test)}}$. A parametrized function $f$, i.e., a neural network, approximates the targets as $\hat{\mathbf{y}} = f(\mathbf{X}; \boldsymbol{\theta})$, where the parameters $\boldsymbol{\theta}$ are trained to minimize a differentiable loss function $\mathcal{L}$ as $\arg\min_{\boldsymbol{\theta}} \mathcal{L}\left(\mathbf{y}^{\text{(Train)}}, f\left(\mathbf{X}^{\text{(Train)}}; \boldsymbol{\theta}\right)\right)$. To generalize into minimizing $\mathcal{L}\left(\mathbf{y}^{\text{(Test)}}, f(\mathbf{X}^{\text{(Test)}}; \boldsymbol{\theta})\right)$, the parameters of $f$ are controlled with a regularization technique $\Omega$ that avoids overfitting to the peculiarities of the training data. With a slight abuse of notation we denote $f\left(\mathbf{X}; \Omega\left(\boldsymbol{\theta}; \boldsymbol{\lambda}\right)\right)$ to be the predictions of the model $f$ whose parameters $\boldsymbol{\theta}$ are optimized under the regime of the regularization

method $\Omega(\cdot; \boldsymbol{\lambda})$, where $\boldsymbol{\lambda} \in \boldsymbol{\Lambda}$ represents the hyperparameters of $\Omega$. The training data is further divided into two subsets as training and validation splits, the later denoted by $\mathbf{X}^{(\text{Val})}, \mathbf{y}^{(\text{Val})}$, such that $\boldsymbol{\lambda}$ can be tuned on the validation loss via the following hyperparameter optimization objective:

$$\boldsymbol{\lambda}^* \in \underset{\boldsymbol{\lambda} \in \boldsymbol{\Lambda}}{\arg\min} \quad \mathcal{L}\left(\mathbf{y}^{(\text{Val})}, f\left(\mathbf{X}^{(\text{Val})}; \boldsymbol{\theta}^*_{\boldsymbol{\lambda}}\right)\right), \tag{1}$$

$$\text{s.t.} \quad \boldsymbol{\theta}^*_{\boldsymbol{\lambda}} \in \underset{\boldsymbol{\theta}}{\arg\min} \quad \mathcal{L}\left(\mathbf{y}^{(\text{Train})}, f(\mathbf{X}^{(\text{Train})}; \Omega\left(\boldsymbol{\theta}; \boldsymbol{\lambda}\right)\right).$$

After finding the optimal (or in practice at least a well-performing) configuration $\boldsymbol{\lambda}^*$, we re-fit $\boldsymbol{\theta}$ on the entire training dataset, i.e., $\mathbf{X}^{(\text{Train})} \cup \mathbf{X}^{(\text{Val})}$ and $\mathbf{y}^{(\text{Train})} \cup \mathbf{y}^{(\text{Val})}$.

While the search for optimal hyperparameters $\boldsymbol{\lambda}$ is an active field of research in the realm of AutoML [23], still the choice of the regularizer $\Omega$ mostly remains an ad-hoc practice, where practitioners select a few combinations among popular regularizers (Dropout, L2, Batch Normalization, etc.). In contrast to prior studies, we hypothesize that the optimal regularizer is a cocktail mixture of a large set of regularization methods, all being simultaneously applied with different strengths (i.e., dataset-specific hyperparameters). Given a set of $K$ regularizers $\left\{(\Omega^{(k)}\left(\cdot; \boldsymbol{\lambda}^{(k)}\right)\right\}_{k=1}^{K} := \left\{\Omega^{(1)}\left(\cdot; \boldsymbol{\lambda}^{(1)}\right), \ldots, \Omega^{(K)}\left(\cdot; \boldsymbol{\lambda}^{(K)}\right)\right\}$, each with its own hyperparameters $\boldsymbol{\lambda}^{(k)} \in \boldsymbol{\Lambda}^{(k)}, \forall k \in \{1, \ldots, K\}$, the problem of finding the optimal cocktail of regularizers is:

$$\boldsymbol{\lambda}^* \in \underset{\boldsymbol{\lambda} := (\boldsymbol{\lambda}^{(1)}, \ldots, \boldsymbol{\lambda}^{(K)}) \in (\boldsymbol{\Lambda}^{(1)}, \ldots, \boldsymbol{\Lambda}^{(K)})}{\arg\min} \mathcal{L}\left(\mathbf{y}^{(\text{Val})}, f\left(\mathbf{X}^{(\text{Val})}; \boldsymbol{\theta}^*_{\boldsymbol{\lambda}}\right)\right) \tag{2}$$

$$\text{s.t.:} \quad \boldsymbol{\theta}^*_{\boldsymbol{\lambda}} \in \underset{\boldsymbol{\theta}}{\arg\min} \quad \mathcal{L}\left(\mathbf{y}^{(\text{Train})}, f\left(\mathbf{X}^{(\text{Train})}; \left\{\Omega^{(k)}\left(\boldsymbol{\theta}, \boldsymbol{\lambda}^{(k)}\right)\right\}_{k=1}^{K}\right)\right)$$

The intuitive interpretation of Equation 2 is searching for the optimal hyperparameters $\boldsymbol{\lambda}$ (i.e., strengths) of the cocktail's regularizers using the validation set, given that the optimal prediction model parameters $\boldsymbol{\theta}$ are trained under the regime of all the regularizers being applied jointly. We stress that, for each regularizer, the hyperparameters $\boldsymbol{\lambda}^{(k)}$ include a conditional hyperparameter controlling whether the $k$-th regularizer is applied or skipped. The best cocktail might comprise only a subset of regularizers.

## 4.2 Cocktail Search Space

To build our regularization cocktails we combine the 13 regularization methods listed in Table 1, which represent the categories of regularizers covered in Section 3. The regularization cocktail's search space with the exact ranges for the selected regularizers' hyperparameters is given in the same table. In total, the optimal cocktail is searched in a space of 19 hyperparameters.

While we can in principle use any hyperparameter optimization method, we decided to use the multi-fidelity Bayesian optimization method BOHB [12] since it achieves strong performance across a wide range of computing budgets by combining Hyperband [33] and Bayesian Optimization [40], and since BOHB can deal with the categorical hyperparameters we use for enabling or disabling regularization techniques and the corresponding conditional structures. Appendix A describes the implementation details for the deployed HPO method. Some of the regularization methods cannot be combined, and we, therefore, introduce the following constraints to the proposed search space: (i) Shake-Shake and Shake-Drop are not simultaneously active since the latter builds on the former; (ii) Only one data augmentation technique out of Mix-Up, Cut-Mix, Cut-Out, and FGSM adversarial learning can be active at once due to a technical limitation of the base library we use [62].

## 5 Experimental Protocol

### 5.1 Experimental Setup and Datasets

We use a large collection of 40 tabular datasets (listed in Table 9 of Appendix D). This includes 31 datasets from the recent open-source OpenML AutoML Benchmark [16][2]. In addition, we added

---

[2]The remaining 8 datasets from that benchmark were too large to run effectively on our cluster.

| Group | Regularizer | Hyperparameter | Type | Range | Conditionality |
|-------|-------------|----------------|------|-------|----------------|
| Implicit | BN | BN-active | Boolean | {True, False} | − |
| | SWA | SWA-active | Boolean | {True, False} | - |
| | LA | LA-active | Boolean | {True, False} | − |
| | | Step size | Continuous | $[0.5, 0.8]$ | LA-active |
| | | Num. steps | Integer | $[5, 10]$ | LA-active |
| W. Decay | WD | WD-active | Boolean | {True, False} | − |
| | | Decay factor | Continuous | $[10^{-5}, 0.1]$ | WD-active |
| Ensemble | DO | DO-active | Boolean | {True, False} | − |
| | | Dropout shape | Nominal | {funnel, long funnel, diamond, hexagon, brick, triangle, stairs} | DO-active |
| | | Drop rate | Continuous | $[0.0, 0.8]$ | DO-active |
| | SE | SE-active | Boolean | {True, False} | - |
| Structural | SC | SC-active | Boolean | {True, False} | − |
| | | MB choice | Nominal | {SS, SD, Standard} | SC-active |
| | SD | Max. probability | Continuous | $[0.0, 1.0]$ | SC-active $\wedge$ MB choice = SD |
| | SS | - | - | - | SC-active $\wedge$ MB choice = SS |
| Augmentation | − | Augment | Nominal | {MU, CM, CO, AT, None} | − |
| | MU | Mix. magnitude | Continuous | $[0.0, 1.0]$ | Augment = MU |
| | CM | Probability | Continuous | $[0.0, 1.0]$ | Augment = CM |
| | CO | Probability | Continuous | $[0.0, 1.0]$ | Augment = CO |
| | | Patch ratio | Continuous | $[0.0, 1.0]$ | Augment = CO |
| | AT | - | - | - | Augment = AT |

Table 1: The configuration space for the regularization cocktail regarding the **explicit regularization hyperparameters** of the methods and the conditional constraints enabling or disabling them. (BN: Batch Normalization, SWA: Stochastic Weight Averaging, LA: Lookahead Optimizer, WD: Weight Decay, DO: Dropout, SE: Snapshot Ensembles, SC: Skip Connection, MB: Multi-branch choice, SD: Shake-Drop, SS: Shake-Shake, MU: Mix-Up, CM: Cut-Mix, CO: Cut-Out, and AT: FGSM Adversarial Learning)

9 popular datasets from UCI [3] and Kaggle that contain roughly 100K+ instances. Our resulting benchmark of 40 datasets includes tabular datasets that represent diverse classification problems, containing between 452 and 416 188 data points, and between 4 and 2 001 features, varying in terms of the number of numerical and categorical features. The datasets are retrieved from the OpenML repository [54] using the OpenML-Python connector [14] and split as 60% training, 20% validation, and 20% testing sets. The data is standardized to have zero mean and unit variance where the statistics for the standardization are calculated on the training split.

We ran all experiments on a CPU cluster, each node of which contains two Intel Xeon E5-2630v4 CPUs with 20 CPU cores each, running at 2.2GHz and a total memory of 128GB. We chose the PyTorch library [43] as a deep learning framework and extended the AutoDL-framework Auto-Pytorch [39, 62] with our implementations for the regularizers of Table 1. We provide the code for our implementation at the following link: `https://github.com/releaunifreiburg/WellTunedSimpleNets`.

To optimally utilize resources, we ran BOHB with 10 workers in parallel, where each worker had access to 2 CPU cores and 12GB of memory, executing one configuration at a time. Taking into account the dimensions $D$ of the considered configuration spaces, we ran BOHB for at most 4 days, or at most $40 \times D$ hyperparameter configurations, whichever came first. During the training phase, each configuration was run for 105 epochs, in accordance with the cosine learning rate annealing with restarts (described in the following subsection). For the sake of studying the effect on more datasets, we only evaluated a single train-val-test split. After the training phase is completed, we report the results of the best hyperparameter configuration found, retrained on the joint train and validation set.

## 5.2 Fixed Architecture and Optimization Hyperparameters

In order to focus exclusively on investigating the effect of regularization we fix the neural architecture to a simple multilayer perceptron (MLP) and also fix some hyperparameters of the general training

procedure. These fixed hyperparameter values, as specified in Table 4 of Appendix B.1, have been tuned for maximizing the performance of an unregularized neural network on our dataset collection (see Table 9 in Appendix D). We use a 9-layer feed-forward neural network with 512 units for each layer, a choice motivated by previous work [42].

Moreover, we set a low learning rate of $10^{-3}$ after performing a grid search for finding the best value across datasets. We use AdamW [36], which implements decoupled weight decay, and cosine annealing with restarts [35] as a learning rate scheduler. Using a learning rate scheduler with restarts helps in our case because we keep a fixed initial learning rate. For the restarts, we use an initial budget of 15 epochs, with a budget multiplier of 2, following published practices [62]. Additionally, since our benchmark includes imbalanced datasets, we use a weighted version of categorical cross-entropy and balanced accuracy [4] as the evaluation metric.

## 5.3 Research Hypotheses and Associated Experiments

**Hypothesis 1:** Regularization cocktails outperform state-of-the-art deep learning architectures on tabular datasets.

**Experiment 1:** We compare our well-regularized MLPs against the recently proposed deep learning architectures Node [46] and TabNet [1]. Additionally, we compare against two versions of AutoGluon Tabular [11], a version that features stacking and a version that additionally includes hyperparameter optimization. Moreover, we add an unregularized version of our MLP for reference, as well as a version of our MLP regularized with Dropout (where the dropout hyperparameters are tuned on every dataset). Lastly, we also compare against self-normalizing neural networks [29] by using the same MLP backbone as with our regularization cocktails.

**Hypothesis 2:** Regularization cocktails outperform Gradient-Boosted Decision Trees (GBDTs), the most commonly used traditional ML method and de-facto state-of-the-art for tabular data.

**Experiment 2:** We compare against three different implementations of GBDT: an implementation from scikit-learn [44] and optimized by Auto-sklearn [13], the popular XGBoost [5], and lastly, the recently proposed CatBoost [47].

**Hypothesis 3:** Regularization cocktails are time-efficient and achieve strong anytime results.

**Experiment 3:** We compare our regularization cocktails against XGBoost over time.

## 5.4 Experimental Setup for the Baselines

All baselines use the same train, validation, and test splits, the same seed, and the same HPO resources and constraints as for our automatically-constructed regularization cocktails (4 days on 20 CPU cores with 128GB of memory). After finding the best incumbent configuration, the baselines are refitted on the union of the training and validation sets and evaluated on the test set. The baselines consist of two recent neural architectures, two versions of AutoGluon Tabular with neural networks, and three implementations of GBDT, as follows:

**TabNet:** This library does not provide an HPO algorithm by default; therefore, we also used BOHB for this search space, with the hyperparameter value ranges recommended by the authors [1].

**Node:** This library does not offer an HPO algorithm by default. We performed a grid search among the hyperparameter value ranges as proposed by the authors [46]; however, we faced multiple memory and runtime issues in running the code. To overcome these issues we used the default hyperparameters the authors used in their public implementation.

**AutoGluon Tabular:** This library constructs stacked ensembles with bagging among diverse neural network architectures having various kinds of regularization [11]. The training of the stacking ensemble of neural networks and its hyperparameter tuning are integrated into the library. Hyperparameter optimization (HPO) is deactivated by default to give more resources to stacking, but here we study AutoGluon based on either stacking or HPO (and HPO actually performs somewhat better). While AutoGluon Tabular by default uses a broad range of traditional ML techniques, here, in order to study it as a "pure" deep learning method, we restrict it to only use neural networks as base learners.

**ASK-GBDT:** The GBDT implementation of scikit-learn offered by Auto-sklearn [13] uses SMAC for HPO, and we used the default hyperparameter search space given by the library.

**XGBoost:** The original library [5] does not incorporate an HPO algorithm by default, so we used BOHB for its HPO. We defined a search space for XGBoost's hyperparameters following the best practices by the community; we describe this in the Appendix B.2.

**CatBoost:** Like for XGBoost, the original library [47] does not incorporate an HPO algorithm, so we used BOHB for its HPO, with the hyperparameter search space recommended by the authors.

For in-depth details about the different baseline configurations with the exact hyperparameter search spaces, please refer to Appendix B.2.

| Dataset | #Ins./#Feat. | MLP | MLP+D | XGB. | ASK-G. | TabN. | Node | AutoGL. S | MLP+C |
|---|---|---|---|---|---|---|---|---|---|
| anneal | 898 / 39 | 84.131 | 86.916 | 85.416 | **90.000** | 84.248 | 20.000 | 80.000 | 89.270 |
| kr-vs-kp | 3196 / 37 | 99.701 | **99.850** | **99.850** | **99.850** | 93.250 | 97.264 | 99.687 | **99.850** |
| arrhythmia | 452 / 280 | 37.991 | 38.704 | 48.779 | 46.850 | 43.562 | N/A | 48.934 | **61.461** |
| mfeat. | 2000 / 217 | 97.750 | **98.000** | **98.000** | 97.500 | 97.250 | 97.250 | **98.000** | **98.000** |
| credit-g | 1000 / 21 | 69.405 | 68.095 | 68.929 | 71.191 | 61.190 | 73.095 | 69.643 | **74.643** |
| vehicle | 846 / 19 | 83.766 | 82.603 | 74.973 | 80.165 | 79.654 | 75.541 | **83.793** | 82.576 |
| kc1 | 2109 / 22 | 70.274 | 72.980 | 66.846 | 63.353 | 52.517 | 55.803 | 67.270 | **74.381** |
| adult | 48842 / 15 | 76.893 | 78.520 | 79.824 | 79.830 | 77.155 | 78.168 | 80.557 | **82.443** |
| walking. | 149332 / 5 | 60.997 | 63.754 | 61.616 | 62.764 | 56.801 | N/A | 60.800 | **63.923** |
| phoneme | 5404 / 6 | 87.514 | **88.387** | 87.972 | 88.341 | 86.824 | 82.720 | 83.943 | 86.619 |
| skin-seg. | 245057 / 4 | 99.971 | 99.962 | 99.968 | 99.967 | 99.961 | N/A | **99.973** | 99.953 |
| ldpa | 164860 / 8 | 62.831 | 67.035 | **99.008** | 68.947 | 54.815 | N/A | 53.023 | 68.107 |
| nomao | 34465 / 119 | 95.917 | 96.232 | 96.872 | **97.217** | 95.425 | 96.217 | 96.420 | 96.826 |
| cnae | 1080 / 857 | 87.500 | 90.741 | 94.907 | 93.519 | 89.352 | **96.759** | 92.593 | 95.833 |
| blood. | 748 / 5 | 67.836 | **68.421** | 62.281 | 64.985 | 64.327 | 50.000 | 67.251 | 67.617 |
| bank. | 45211 / 17 | 78.076 | 83.145 | 72.658 | 72.283 | 70.639 | 74.607 | 79.483 | **85.993** |
| connect. | 67557 / 43 | 73.627 | 76.345 | 72.374 | 72.645 | 72.045 | N/A | 75.622 | **80.073** |
| shuttle | 58000 / 10 | 99.475 | 99.892 | 98.563 | 98.571 | 88.017 | 42.805 | 83.433 | **99.948** |
| higgs | 98050 / 29 | 67.752 | 66.873 | 72.944 | 72.926 | 72.036 | N/A | **73.798** | 73.546 |
| australian | 690 / 15 | 86.268 | 86.268 | **89.717** | 88.589 | 85.278 | 83.468 | 88.248 | 87.088 |
| car | 1728 / 7 | 97.442 | 99.690 | 92.376 | **100.000** | 98.701 | 46.119 | 99.675 | 99.587 |
| segment | 2310 / 20 | **94.805** | 94.589 | 93.723 | 93.074 | 91.775 | 90.043 | 91.991 | 93.723 |
| fashion. | 70000 / 785 | 90.464 | 90.507 | 91.243 | 90.457 | 89.793 | N/A | 91.336 | **91.950** |
| jungle. | 44819 / 7 | 97.061 | 97.237 | 87.325 | 83.070 | 73.425 | N/A | 93.017 | **97.471** |
| numerai | 96320 / 22 | 50.262 | 50.301 | 52.363 | 52.421 | 51.599 | 52.364 | 51.706 | **52.668** |
| devnagari | 92000 / 1025 | 96.125 | 97.000 | 93.310 | 77.897 | 94.179 | N/A | 97.734 | **98.370** |
| helena | 65196 / 28 | 16.836 | 23.983 | 21.994 | 21.144 | 19.032 | N/A | 27.115 | **27.701** |
| jannis | 83733 / 55 | 51.505 | 55.118 | 55.225 | 55.593 | 56.214 | N/A | 58.526 | **65.287** |
| volkert | 58310 / 181 | 65.081 | 66.996 | 64.170 | 63.428 | 59.409 | N/A | 70.195 | **71.667** |
| miniboone | 130064 / 51 | 90.639 | 94.099 | 94.024 | 94.137 | 62.173 | N/A | **94.978** | 94.015 |
| apsfailure | 76000 / 171 | 87.759 | 91.194 | 88.825 | 91.797 | 51.444 | N/A | 88.890 | **92.535** |
| christine | 5418 / 1637 | 70.941 | 70.756 | **74.815** | 74.447 | 69.649 | 73.247 | 74.170 | 74.262 |
| dilbert | 10000 / 2001 | 96.930 | 96.733 | **99.106** | 98.704 | 97.608 | N/A | 98.758 | 99.049 |
| fabert | 8237 / 801 | 63.707 | 64.814 | 70.098 | **70.120** | 62.277 | 66.097 | 68.142 | 69.183 |
| jasmine | 2984 / 145 | 78.048 | 76.211 | **80.546** | 78.878 | 76.690 | 80.053 | 80.046 | 79.217 |
| sylvine | 5124 / 21 | 93.070 | 93.363 | **95.509** | 95.119 | 83.595 | 93.852 | 93.753 | 94.045 |
| dionis | 416188 / 61 | 91.905 | 92.724 | 91.222 | 74.620 | 83.960 | N/A | **94.127** | 94.010 |
| aloi | 108000 / 129 | 92.331 | 93.852 | 95.338 | 13.534 | 93.589 | N/A | **97.423** | 97.175 |
| ccfraud | 284807 / 31 | 50.000 | 50.000 | 90.303 | 92.514 | 85.705 | N/A | 91.831 | **92.531** |
| clickpred. | 399482 / 12 | 63.125 | **64.367** | 58.361 | 58.201 | 50.163 | N/A | 54.410 | 64.280 |
| **Wins/Losses/Ties** | **MLP+C** vs … | 35/5/0 | 30/8/2 | 26/12/2 | 29/11/0 | 38/2/0 | 19/2/0 | 30/9/1 | - |
| **Wilcoxon $p$-value** | **MLP+C** vs … | $5.3 \times 10^{-7}$ | $8.9 \times 10^{-6}$ | $6 \times 10^{-4}$ | $2.8 \times 10^{-4}$ | $4.5 \times 10^{-8}$ | $8.2 \times 10^{-8}$ | $4 \times 10^{-5}$ | - |

Table 2: Comparison of well-regularized MLPs vs. other methods in terms of balanced accuracy. N/A values indicate a failure due to exceeding the cluster's memory (24GB per process) or runtime limits (4 days). The acronyms stand for MLP+D: MLP with Dropout, XGB.: XGBoost, ASK-G.: GBDT by Auto-sklearn, AutoGL. S: Autogluon with stacking enabled, TabN.: TabNet and MLP+C: our MLP regularized by cocktails.

## 6 Experimental Results

We present the comparative results of our MLPs regularized with the proposed regularization cocktails (MLP+C) against ten baselines (descriptions in Section 5.4): *(a)* two state-of-the-art architectures (NODE, TabN.); *(b)* two AutoGluon Tabular variants with neural networks that features stacking (AutoGL. S) and additionally HPO (AutoGL. HPO); *(c)* three Gradient-Boosted Decision Tree

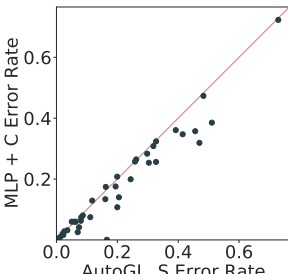 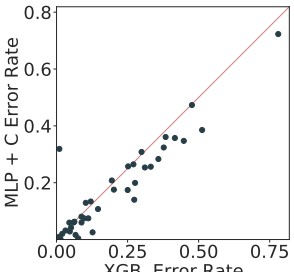 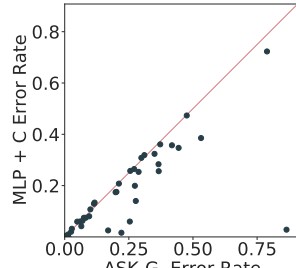

Figure 1: Comparison of our proposed dataset-specific cocktail (MLP+C) against the top three baselines. Each dot in the plot represents a dataset, the y-axis our method's errors and the x-axis the baselines' errors.

(GBDT) implementations (XGB., ASK-G., and CatBoost); *(d)* as well as three reference MLPs (unregularized (MLP), and regularized with Dropout (MLP+D) [49] or SELU (MLP+SELU) [30]).

Table 2 shows the comparison against a subset of the baselines, while the full detailed results involving all the remaining baselines are located in Table 13 in the appendix. It is worth re-emphasizing that the hyperparameters of all the presented baselines (except the unregularized MLP, which has no hyperparameters and Au-toGL. S) are carefully tuned on a validation set as detailed in Section 5 and the appendices referenced therein. The table entries represent the test sets' balanced accuracies achieved over the described large-scale collection of 40 datasets. Figure 1 visualizes the results showing substantial improvements for our method.

To assess the statistical significance, we analyze the ranks of the classification accuracies across the 40 datasets. We use the Critical Difference (CD) diagram of the ranks based on the Wilcoxon significance test, a standard metric for comparing classifiers across multiple datasets [9]. The overall empirical comparison of the elaborated methods is given in Figure 2. The analysis of neural network baselines in Subplot 2a reveals a clear statistical significance of the regularization cocktails against the other methods. Apart from AutoGluon (both versions), the other neural architectures are not competitive even against an MLP regularized only with Dropout and optimized with our standard, fixed training pipeline of Adam with cosine annealing. To be even fairer to the weaker

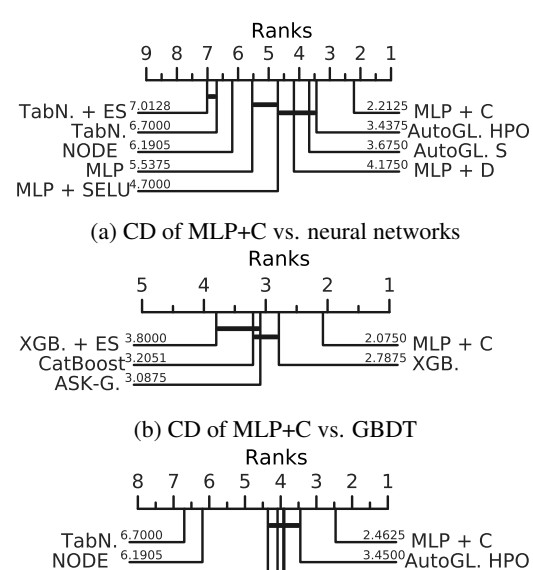

(a) CD of MLP+C vs. neural networks

(b) CD of MLP+C vs. GBDT

(c) CD of MLP+C vs. all baselines

Figure 2: **Critical difference diagrams** with a Wilcoxon significance analysis on 40 datasets. Connected ranks via a bold bar indicate that performances are not significantly different ($p > 0.05$).

baselines (TabNet and Node) we tried boosting them by adding early stopping (indicated with "+ES"), but their rank did not improve. Overall, the large-scale experimental analysis shows that Hypothesis 1 in Section 5.3 is validated: **well-regularized simple deep MLPs outperform specialized neural architectures**.

Next, we analyze the empirical significance of our well-regularized MLPs against the GBDT implementations in Figure 2b. The results show that our MLPs outperform all three GBDT variants (XGBoost, auto-sklearn, and CatBoost) with a statistically significant margin. We added early stopping ("+ES") to XGBoost, but it did not improve its performance. Among the GBDT implementations, XGBoost without early stopping has a non-significant margin over the GBDT version of auto-sklearn

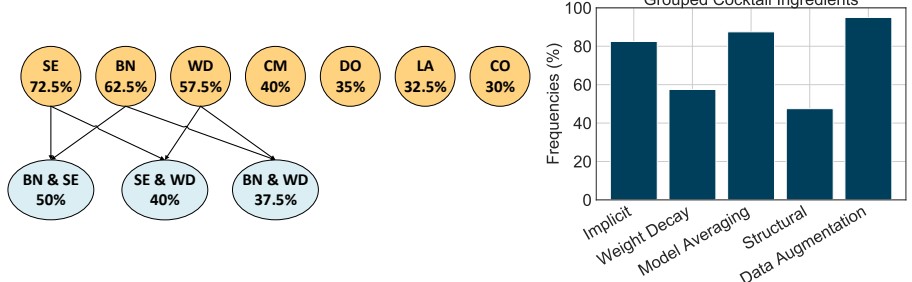

Figure 3: **Left**: Cocktail ingredients occurring in at least 30% of the datasets. **Right**: Clustered histogram (union of member occurrences) with the acronyms from Table 1. Implicit: {BN, LA, SWA}, M. Averaging: {DO, SE}, Structural: {SC, SS, SD}, D. Augmentation: {MU, CM, CO, AT}.

as well as CatBoost. We conclude that **well-regularized simple deep MLPs outperform GBDT**, which validates Hypothesis 2 in Section 5.3.

The final cumulative comparison in Figure 2c provides a further result: none of the specialized previous deep learning methods (TabNet, NODE, AutoGluon Tabular) outperforms GBDT significantly. To the best of our awareness, this paper is therefore the first to demonstrate that neural networks beat GBDT with a statistically significant margin over a large-scale experimental protocol that conducts a thorough hyperparameter optimization for all methods.

Figure 3 provides a further analysis on the most prominent regularizers of the MLP cocktails, based on the frequency with which our HPO procedure selected the various regularization methods for each dataset's cocktail. In the left plot, we show the frequent individual regularizers, while in the right plot the frequencies are grouped by types of regularizers. The grouping reveals that a cocktail for each dataset often has at least one ingredient from every regularization family (detailed in Section 3), highlighting the need for jointly applying diverse regularization methods.

| Time (Hours) | Wins | Ties | Losses | p-value |
|---|---|---|---|---|
| 0.25 | 21 | 1 | 17 | 0.8561 |
| 0.5 | 25 | 1 | 13 | 0.0145 |
| 1 | 24 | 1 | 14 | 0.0120 |
| 2 | 27 | 1 | 11 | 0.0006 |
| 4 | 28 | 1 | 11 | 0.0006 |
| 8 | 28 | 1 | 11 | 0.0004 |
| 16 | 28 | 1 | 11 | 0.0005 |
| 32 | 29 | 1 | 10 | 0.0003 |
| 64 | 30 | 1 | 9 | 0.0002 |
| 96 | 30 | 1 | 9 | 0.0002 |

Table 3: Comparing the cocktails and XGBoost over different HPO budgets. The statistics are based on the test performance of the incumbent configurations over all the benchmark datasets.

Lastly, Table 3 shows the efficiency of our regularization cocktails compared to XGBoost over increasing HPO budgets. The descriptive statistics are calculated from the hyperparameter configurations with the best validation performance for all datasets during the HPO search, however, taking their respective test performances for comparison. A dataset is considered in the comparison only if the HPO procedure has managed to evaluate at least one hyperparameter configuration for the cocktail or baseline. As the table shows, our regularization cocktails achieve a better performance in only 15 minutes for the majority of datasets. After 30 minutes of HPO time, regularization cocktails are statistically significantly better than XGBoost. As more time is invested, the performance gap with XGBoost increases, and the results get even more significant; this is further visualized in the ranking plot over time in Figure 4. Based on these results, we conclude that **regularization cocktails are time-efficient and achieve strong anytime results**, which validates Hypothesis 3 in Section 5.3.

## 7 Conclusion

**Summary.** Focusing on the important domain of tabular datasets, this paper studied improvements to deep learning (DL) by better regularization techniques. We presented *regularization cocktails*, per-dataset-optimized combinations of many regularization techniques, and demonstrated that these improve the performance of even simple neural networks enough to substantially and significantly

surpass XGBoost, the current state-of-the-art method for tabular datasets. We conducted a large-scale experiment involving 13 regularization methods and 40 datasets and empirically showed that (i) modern DL regularization methods developed in the context of raw data (e.g., vision, speech, text) substantially improve the performance of deep neural networks on tabular data; (ii) regularization cocktails significantly outperform recent neural networks architectures, and most importantly iii) regularization cocktails outperform GBDT on tabular datasets.

**Limitations.** To comprehensively study basic principles, we have chosen an empirical evaluation that has many limitations. We only studied classification, not regression. We only used somewhat balanced datasets (the ratio of the minority class and the majority class is above 0.05). We did not study the regimes of extremely few or extremely many data points (our smallest data set contained 452 data points, our largest 416 188 data points). We also did not study datasets with extreme outliers, missing labels, semi-supervised data, streaming data, and many more modalities in which tabular data arises. An important point worth noticing is that the recent neural network architectures (Section 5.4) could also benefit from our regularization cocktails, but integrating the regularizers into these baseline libraries requires considerable coding efforts.

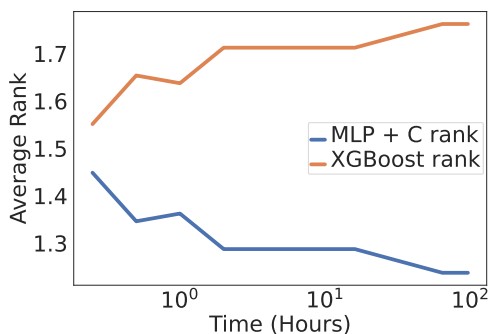

Figure 4: Ranking plot comparing XGBoost and regularization cocktails over time.

**Future Work.** This work opens up the door for a wealth of exciting follow-up research. Firstly, the per-dataset optimization of regularization cocktails may be substantially sped up by using meta-learning across datasets [53]. Secondly, as we have used a fixed neural architecture, our method's performance may be further improved by using joint architecture and hyperparameter optimization. Thirdly, regularization cocktails should also be tested under all the data modalities under "Limitations" above. In addition, it would be interesting to validate the gain of integrating our well-regularized MLPs into modern AutoML libraries, by combining them with enhanced feature preprocessing and ensembling.

**Take-away.** *Even simple neural networks can achieve competitive classification accuracies on tabular datasets when they are well regularized, using dataset-specific regularization cocktails found via standard hyperparameter optimization.*

## Societal Implications

Enabling neural networks to advance the state-of-the-art on tabular datasets may open up a garden of delights in many crucial applications, such as climate science, medicine, manufacturing, and recommender systems. In addition, the proposed networks can serve as a backbone for applications of data science for social good, such as the realm of fair machine learning where the associated data are naturally in a tabular form. However, there are also potential disadvantages in advancing deep learning for tabular data. In particular, even though complex GBDT ensembles are also hard to interpret, simpler traditional ML methods are much more interpretable than deep neural networks; we therefore encourage research on interpretable deep learning on tabular data.

## Acknowledgements

We acknowledge funding by the Robert Bosch GmbH, by the Eva Mayr-Stihl Foundation, the MWK of the German state of Baden-Württemberg, the BrainLinks-BrainTools CoE, the European Research Council (ERC) under the European Union's Horizon 2020 programme, grant no. 716721, and the German Federal Ministry of Education and Research (BMBF, grant RenormalizedFlows 01IS19077C). The authors acknowledge support by the state of Baden-Württemberg through bwHPC and the German Research Foundation (DFG) through grant no INST 39/963-1 FUGG.

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
