## A  Description of BOHB

BOHB [12] is a hyperparameter optimization algorithm that extends Hyperband [33] by sampling from a model instead of sampling randomly from the hyperparameter search space.

Initially, BOHB performs random search and favors exploration. As it iterates and gets more observations, it builds models over different fidelities and trades off exploration with exploitation to avoid converging in bad regions of the search space. BOHB samples from the model of the highest fidelity with a probability $p$ and with $1 - p$ from random. A model is built for a fidelity only when enough observations exist for that fidelity; by default, this limit is set to equal $S + 1$ observations, where $S$ is the dimensionality of the search space.

We used the original implementation of BOHB in HPBandSter[3] [12]. For simplicity, we only used a single fidelity level for BOHB, effectively using it as a blackbox optimization algorithm; we believe that future work should revisit this choice.

## B  Configuration Spaces

### B.1  Method implicit search space

| Category | Hyperparameter | Type | Range |
|---|---|---|---|
| Cosine Annealing | Iterations multiplier | Continuous | {2.0} |
| | Max. iterations | Integer | {15} |
| Network | Activation | Nominal | {ReLU} |
| | Bias initialization | Nominal | {Yes} |
| | Blocks in a group | Integer | {2} |
| | Embeddings | Nominal | {One-Hot encoding} |
| | Number of groups | Integer | {2} |
| | Resnet shape | Nominal | {Brick} |
| | Type | Nominal | {Shaped-Resnet} |
| | Units in a layer | Integer | {512} |
| Preprocessing | Preprocessor | Nominal | {None} |
| Training | Batch size | Integer | {128} |
| | Imputation | Nominal | {Median} |
| | Initialization method | Nominal | {Default} |
| | Learning rate | Continuous | $\{10^{-3}\}$ |
| | Loss module | Nominal | {Weighted Cross-Entropy} |
| | Normalization strategy | Nominal | {Standardize} |
| | Optimizer | Nominal | {AdamW} |
| | Scheduler | Nominal | {COS} |
| | Seed | Integer | {11} |

Table 4: The configuration space of the training and model architecture hyperparameters. All these hyperparameters only have one value in their range, meaning they are fixed.

Table 4 presents the network architecture and the training pipeline choices used in all our experiments for the individual regularizers and for the regularization cocktails.

### B.2  Benchmark search space

For the experiments conducted in our work, we set up the search space and the individual configurations of the state-of-the-art competitors used for the comparison as follows:

**Auto-Sklearn.** The estimator is restricted to only include GBDT, for the sake of fully comparing against the algorithm as a baseline. We do not activate any preprocessing since our regularization cocktails also do not make use of preprocessing algorithms in the pipeline. The time left is always selected based on the time it took BOHB to find the hyperparameter with the best validation accuracy from the start of the hyperparameter optimization phase. The ensemble size is kept to 1 since our method only uses models from one training run, not multiple ones. The seed is set to 11 as it was set in the experiments with the regularization cocktail, to obtain the same data splits. To keep the comparison fair, there is no warm start for the initial configurations with meta-learning, since our method also does not make use of meta-learning. Lastly, the number of parallel workers is set to 10, to match the parallel resources that were given to the experiment with the regularization cocktails. The search space of the hyperparameters is left to the default search space offered by Auto-Sklearn which is shown in Table 5. We use version 0.10.0 of the library.

---

[3] `https://github.com/automl/HpBandSter`

| Hyperparameter | Type | Range |
|---|---|---|
| $early\_stopping$ | Nominal | {Off, Train, Valid} |
| $l_2\_regularization$ | Continuous | $[1e-10, 1]$ |
| $learning\_rate$ | Continuous | $[0.01, 1]$ |
| $max\_leaf\_nodes$ | Integer | $[3, 2047]$ |
| $min\_samples\_leaf$ | Integer | $[1, 200]$ |
| $nr\_iterations\_no\_change$ | Integer | $[1, 20]$ |
| $validation\_fraction$ | Continuous | $[0.01, 0.4]$ |

Table 5: The search space of the training and model hyperparameters for the gradient boosting estimator of the Auto-Sklearn tool.

**XGBoost.** To have a well-performing configuration space for XGBoost we augmented the default configuration spaces previously used in Auto-Sklearn[4] with further recommended hyperparameters and ranges from Amazon[5]. In Table 6, we present a refined version of the configuration space that achieves a better performance on the benchmark. We would like to note that we did not apply One-Hot encoding to the categorical features for the experiment, since we observed better overall results when the categorical features were ordinal encoded. Nevertheless, in Table 13 we ablate the choice of encoding (+ENC) as a categorical hyperparameter (one-hot vs. ordinal) and also early stopping (+ES). When early stopping is activated, the $num\_rounds$ is increased to 4000. All the results can be found in Appendix D.

| Hyperparameter | Type | Range | Log scale |
|---|---|---|---|
| $eta$ | Continuous | $[0.001, 1]$ | ✓ |
| $lambda$ | Continuous | $[1e-10, 1]$ | ✓ |
| $alpha$ | Continuous | $[1e-10, 1]$ | ✓ |
| $num\_round$ | Integer | $[1, 1000]$ | - |
| $gamma$ | Continuous | $[0.1, 1]$ | ✓ |
| $colsample\_bylevel$ | Continuous | $[0.1, 1]$ | - |
| $colsample\_bynode$ | Continuous | $[0.1, 1]$ | - |
| $colsample\_bytree$ | Continuous | $[0.5, 1]$ | - |
| $max\_depth$ | Integer | $[1, 20]$ | - |
| $max\_delta\_step$ | Integer | $[0, 10]$ | - |
| $min\_child\_weight$ | Continuous | $[0.1, 20]$ | ✓ |
| $subsample$ | Continuous | $[0.01, 1]$ | - |

Table 6: The hyperparameter search space for the XGBoost library.

**TabNet.** For the search space of the TabNet model, we used the default hyperparameter ranges suggested by the authors which were found to perform best in their experiments.

For our experiments with the TabNet and XGBoost models, we also used BOHB for hyperparameter tuning, using the same parallel resources and limiting conditions as for our regularization cocktail. In the above search spaces for the experiments with the XGBoost and TabNet models, we did not include early stopping; however, we did actually run experiments with early stopping for both models,

---

[4] `https://github.com/automl/auto-sklearn/blob/v.0.4.2/autosklearn/pipeline/components/classification/xgradient_boosting.py`

[5] `https://docs.aws.amazon.com/sagemaker/latest/dg/xgboost-tuning.html` We reduced the size of some ranges since the ranges given at this website were too broad and resulted in poor performance.

| Hyperparameter | Type | Values |
|---|---|---|
| $n_a$ | Integer | $\{8, 16, 24, 32, 64, 128\}$ |
| $learning\_rate$ | Continuous | $\{0.005, 0.01, 0.02, 0.025\}$ |
| $gamma$ | Continuous | $\{1.0, 1.2, 1.5, 2.0\}$ |
| $n_{steps}$ | Integer | $\{3, 4, 5, 6, 7, 8, 9, 10\}$ |
| $\lambda_{sparse}$ | Continuous | $\{0, 0.000001, 0.0001, 0.001, 0.01, 0.1\}$ |
| $batch\_size$ | Integer | $\{256, 512, 1024, 2048, 4096, 8192, 16384, 32768\}$ |
| $virtual\_batch\_size$ | Integer | $\{256, 512, 1024, 2048, 4096\}$ |
| $decay\_rate$ | Continuous | $\{0.4, 0.8, 0.9, 0.95\}$ |
| $decay\_iterations$ | Integer | $\{500, 2000, 8000, 10000, 20000\}$ |
| $momentum$ | Continuous | $\{0.6, 0.7, 0.8, 0.9, 0.95, 0.98\}$ |

Table 7: The hyperparameter search space for TabNet.

and results did not improve. Lastly, for both experiments, we imputed missing values with the most frequent strategy (the implementation we used did not accept the median strategy for categorical value imputation).

**AutoGluon.** The library is configured to construct stacked ensembles with bagging among diverse neural network architectures having various kinds of regularization with the $preset = $ **'Best Quality'** to achieve the best predictive accuracy. Furthermore, we used the same seed as for our MLPs with regularization cocktails to obtain the same dataset splits. We allowed AutoGluon to make use of early stopping and additionally, we allowed feature preprocessing since different feature preprocessing techniques are embedded in different model types, to allow for better overall performance. For all the other training and hyperparameter settings, we used the library's default[6] following the explicit recommendation of the authors on the efficacy of their proposed stacking without needing any HPO [11].

We also investigate using HPO with AutoGluon and compare the results against the version that uses stacking; the full detailed results can be found in Appendix D.

**NODE.** For our experiments with NODE we used the official implementation[7]. In our initial experiment iterations, we used the search space that was proposed by the authors [46]. However, evaluating the search space proposed is infeasible, since the memory and run-time requirements of the experiments are very high and cannot be satisfied within our cluster constraints. The high run-time and memory issues are also noted by the authors in the official implementation.

To alleviate these problems, we used the default configuration suggested by the authors in the examples, where $num\_layers = 2$, $total\_tree\_count = 1024$ and $tree\_depth = 6$. Lastly, we use the same seed as for our experiment with the regularization cocktails to obtain the same data splits.

**CatBoost.** We use version 0.26 of the official library and we use the hyperparameter search space that is recommended by the authors [47], provided in Table 8.

## C Plots

### C.1 Regularization Cocktail Performance

To investigate the performance of our formulation, we compare plain MLPs regularized with only one individual regularization technique at a time against the dataset-specific regularization cocktails. The hyperparameters for all methods are tuned on the validation set and the best configuration is refitted on the full training set. In Figure 5, we present the results of each pairwise comparison. The results

---

[6]We used version 0.2.0 of the AutoGluon library
[7]https://github.com/Qwicen/node

| Hyperparameter | Type | Range | Log scale |
|---|---|---|---|
| $learning\_rate$ | Continuous | $[e^{-7}, 1]$ | ✓ |
| $random\_strength$ | Integer | $[1, 20]$ | - |
| $one\_hot\_max\_size$ | Integer | $[0, 25]$ | - |
| $num\_round$ | Integer | $[1, 4000]$ | - |
| $l2\_leaf\_reg$ | Continuous | $[1, 10]$ | ✓ |
| $bagging\_temperature$ | Continuous | $[0, 1]$ | - |
| $gradient\_iterations$ | Integer | $[1, 10]$ | - |

Table 8: The hyperparameter search space for the CatBoost library.

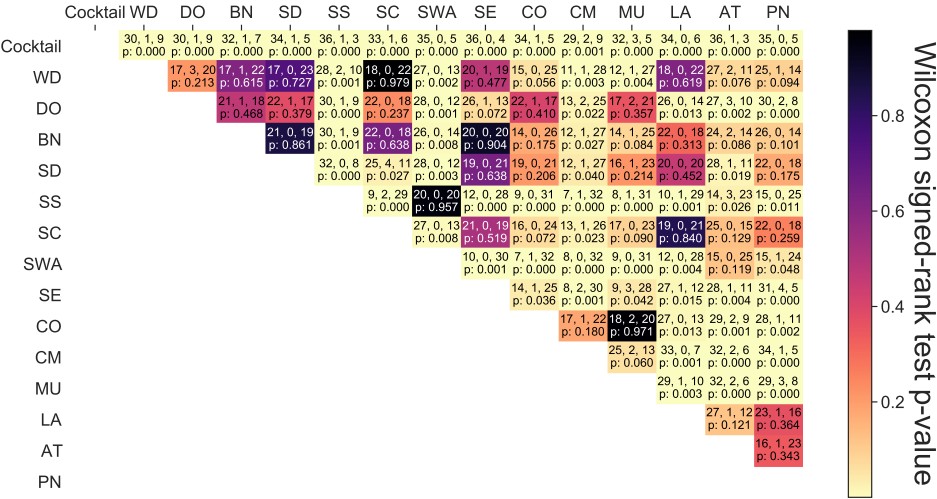

Figure 5: **Pairwise statistical significance and comparison.** For every entry, **the first row** show-cases the wins, draws and losses of the horizontal method with the vertical method on all datasets, calculated on the test set; **the second row** presents the p-value for the statistical significance test.

presented are calculated on the test set after the refit phase is completed on the best hyperparameter configuration. The p-value is generated by performing a Wilcoxon signed-rank test. As can be seen from the results, the regularization cocktail is the only method that has statistically significant improvements compared to all other methods (with a p-value $\leq 0.001$ in all cases). The detailed results for all methods on every dataset are shown in Table 11.

## C.2  Dataset-dependent optimal cocktails

To verify the necessity for dataset-specific regularization cocktails, we initially investigate the best-found hyperparameter configurations to observe the occurrences of individual regularization techniques. In Figure 6, we present the occurrences of every regularization method over all datasets. The occurrences are calculated by analyzing the best-found hyperparameter configuration for each dataset and observing the number of times the regularization method was chosen to be activated by BOHB. As can be seen from Figure 6, there is no regularization method or combination that is always chosen for every dataset.

Additionally, we compare our regularization cocktails against the top-5 frequently chosen regularization techniques and the top-5 best performing regularization techniques. For the top-5 baselines, the regularization techniques are activated and their hyperparameters are tuned on the validation set. The results of the comparison as shown in Table 10 show that the cocktail outperforms both top-5 variants, indicating the need for dataset-specific regularization cocktails.

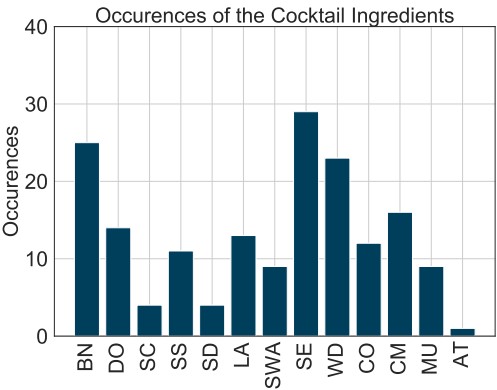

Figure 6: **Frequency of the regularization techniques.** The occurrences of the individual regularization techniques in the best hyperparameter configurations found by the cocktail across 40 datasets.

### C.3 Learning rate as a hyperparameter

In the majority of our experiments, we keep a fixed initial learning rate to investigate in detail the effect of the individual regularization techniques and the regularization cocktails. The learning rate is set to a fixed value that achieves the best results across the chosen benchmark of datasets. To investigate the role and importance of the learning rate in the regularization cocktail performance, we perform an additional experiment, where, the learning rate is an additional hyperparameter that is optimized individually for every dataset. The results as shown in Table 12, indicate that regularization cocktails with a dynamic learning rate outperform the regularization cocktails with a fixed learning rate in 21 out of 40 datasets, tie in 1 and lose in 18. However, the results are not statistically significant with a $p$-value of 0.7 and do not indicate a clear region where the dynamic learning rate helps.

## D Tables

In Table 9, we provide information about the datasets that are considered in our experiments. Concretely, we provide descriptive statistics and the identifiers for every dataset. The identifier (the task id) can be used to download the datasets from OpenML (http://www.openml.org) by using the OpenML-Python connector [14].

Table 10 shows the results for the comparison between the Regularization Cocktail and the Top-5 cocktail variants. The results are calculated on the test set for all datasets, after retraining on the best dataset-specific hyperparameter configuration.

Table 11 provides the results of all our experiments for the plain MLP baseline, the individual regularization methods, and the regularization cocktail. All the results are calculated on the test set, after retraining on the best-found hyperparameter configurations. The evaluation metric used for the performance is balanced accuracy.

Additionally, in Table 12, we provide the results of the regularization cocktails with a fixed learning rate and with the learning rate being a hyperparameter optimized for every dataset.

Lastly, in Table 13, we present the remaining baselines from our experiments/ablations and their final performances on every dataset. The results show the test set performances after the incumbent configuration is refit.

| Task Id | Dataset Name | Number of Instances | Number of Features | Majority Class Percentage | Minority Class Percentage |
|---|---|---|---|---|---|
| 233090 | anneal | 898 | 39 | 76.17 | 0.89 |
| 233091 | kr-vs-kp | 3196 | 37 | 52.22 | 47.78 |
| 233092 | arrhythmia | 452 | 280 | 54.20 | 0.44 |
| 233093 | mfeat-factors | 2000 | 217 | 10.00 | 10.00 |
| 233088 | credit-g | 1000 | 21 | 70.00 | 30.00 |
| 233094 | vehicle | 846 | 19 | 25.77 | 23.52 |
| 233096 | kc1 | 2109 | 22 | 84.54 | 15.46 |
| 233099 | adult | 48842 | 15 | 76.07 | 23.93 |
| 233102 | walking-activity | 149332 | 5 | 14.73 | 0.61 |
| 233103 | phoneme | 5404 | 6 | 70.65 | 29.35 |
| 233104 | skin-segmentation | 245057 | 4 | 79.25 | 20.75 |
| 233106 | ldpa | 164860 | 8 | 33.05 | 0.84 |
| 233107 | nomao | 34465 | 119 | 71.44 | 28.56 |
| 233108 | cnae-9 | 1080 | 857 | 11.11 | 11.11 |
| 233109 | blood-transfusion | 748 | 5 | 76.20 | 23.80 |
| 233110 | bank-marketing | 45211 | 17 | 88.30 | 11.70 |
| 233112 | connect-4 | 67557 | 43 | 65.83 | 9.55 |
| 233113 | shuttle | 58000 | 10 | 78.60 | 0.02 |
| 233114 | higgs | 98050 | 29 | 52.86 | 47.14 |
| 233115 | Australian | 690 | 15 | 55.51 | 44.49 |
| 233116 | car | 1728 | 7 | 70.02 | 3.76 |
| 233117 | segment | 2310 | 20 | 14.29 | 14.29 |
| 233118 | Fashion-MNIST | 70000 | 785 | 10.00 | 10.00 |
| 233119 | Jungle-Chess-2pcs | 44819 | 7 | 51.46 | 9.67 |
| 233120 | numerai28.6 | 96320 | 22 | 50.52 | 49.48 |
| 233121 | Devnagari-Script | 92000 | 1025 | 2.17 | 2.17 |
| 233122 | helena | 65196 | 28 | 6.14 | 0.17 |
| 233123 | jannis | 83733 | 55 | 46.01 | 2.01 |
| 233124 | volkert | 58310 | 181 | 21.96 | 2.33 |
| 233126 | MiniBooNE | 130064 | 51 | 71.94 | 28.06 |
| 233130 | APSFailure | 76000 | 171 | 98.19 | 1.81 |
| 233131 | christine | 5418 | 1637 | 50.00 | 50.00 |
| 233132 | dilbert | 10000 | 2001 | 20.49 | 19.13 |
| 233133 | fabert | 8237 | 801 | 23.39 | 6.09 |
| 233134 | jasmine | 2984 | 145 | 50.00 | 50.00 |
| 233135 | sylvine | 5124 | 21 | 50.00 | 50.00 |
| 233137 | dionis | 416188 | 61 | 0.59 | 0.21 |
| 233142 | aloi | 108000 | 129 | 0.10 | 0.10 |
| 233143 | C.C.FraudD. | 284807 | 31 | 99.83 | 0.17 |
| 233146 | Click prediction | 399482 | 12 | 83.21 | 16.79 |

Table 9: **Datasets.** The collection of datasets used in our experiments, combined with detailed information for each dataset.

| Task Id | Cockt. | Top-5 F | Top-5 R | Task Id | Cockt. | Top-5 F | Top-5 R | Task Id | Cockt. | Top-5 F | Top-5 R |
|---|---|---|---|---|---|---|---|---|---|---|---|
| 233090 | 89.27 | **89.71** | 88.54 | 233091 | **99.85** | **99.85** | 98.20 | 233092 | **61.46** | 59.94 | 57.21 |
| 233093 | 98.00 | **98.75** | **98.75** | 233088 | 74.64 | 71.43 | **74.76** | 233094 | **82.58** | 82.01 | 80.33 |
| 233096 | 74.38 | **78.03** | 73.96 | 233099 | **82.44** | 82.35 | 82.24 | 233102 | **63.92** | 62.21 | 54.10 |
| 233107 | **96.83** | 96.67 | 96.59 | 233108 | 95.83 | 95.83 | 95.83 | 233109 | 67.62 | 67.32 | **68.20** |
| 233110 | 85.99 | **86.35** | 86.06 | 233112 | **80.07** | 79.57 | 77.49 | 233113 | **99.95** | 97.95 | 85.34 |
| 233114 | **73.55** | 73.25 | 72.06 | 233115 | 87.09 | **88.11** | 87.60 | 233116 | 99.59 | **100.00** | 98.20 |
| 233117 | 93.72 | **93.94** | 90.69 | 233118 | **91.95** | 91.83 | 91.59 | 233119 | **97.47** | 92.66 | 85.53 |
| 233120 | **52.67** | 52.49 | 51.70 | 233121 | 98.37 | **98.41** | 96.93 | 233122 | 27.70 | **28.82** | 28.09 |
| 233123 | **65.29** | 65.13 | 62.11 | 233124 | **71.67** | 70.87 | 66.06 | 233126 | **94.02** | 88.13 | 93.16 |
| 233130 | 92.53 | **96.24** | 95.89 | 233131 | 74.26 | 71.86 | **74.63** | 233132 | **99.05** | 98.95 | 98.55 |
| 233133 | **69.18** | 68.75 | 69.03 | 233134 | **79.22** | 78.21 | 77.71 | 233135 | 94.05 | **94.43** | 93.95 |
| 233137 | 94.01 | **94.33** | 92.43 | 233142 | **97.17** | 97.06 | 96.06 | | | | |
| 233146 | 64.28 | **64.53** | 63.28 | 233143 | 92.53 | 92.13 | **92.59** | | | | |

Table 10: **Top-5 baselines.** The test set performance for the Regularization Cocktail against the Top-5 Most Frequent (Top-5 F) and the Top-5 Highest Ranks (Top-5 R) baselines.

233104 | 99.95 | **99.96** | 99.85; 233106 | 68.11 | **68.81** | 55.45.

| Task Id | PN | BN | LA | SE | SWA | SC | AT | SS | SD | MU | CO | CM | WD | DO | Cocktail |
|---------|----|----|----|----|-----|----|----|----|----|----|----|----|----|----|----------|
| 233090 | 84.13 | 86.78 | 83.99 | 86.48 | 87.96 | 87.21 | 86.92 | 84.28 | 87.21 | **89.27** | 85.60 | 86.77 | 87.06 | 86.92 | **89.27** |
| 233091 | 99.70 | 99.85 | 99.70 | 99.70 | 99.55 | **100.00** | 99.85 | 99.85 | 99.69 | 99.85 | 99.55 | 99.85 | 99.85 | 99.85 | 99.85 |
| 233092 | 37.99 | 41.91 | 36.14 | 37.31 | 25.94 | 53.42 | 38.79 | 55.61 | 53.26 | 42.19 | 32.48 | 42.22 | 35.76 | 38.70 | **61.46** |
| 233093 | 97.75 | **98.50** | 96.00 | 97.75 | 69.25 | 98.25 | 97.25 | 97.25 | 98.25 | 98.00 | 98.00 | 97.75 | 98.00 | 98.00 | 98.00 |
| 233088 | 69.40 | 68.69 | 70.83 | 69.76 | 69.40 | 66.43 | 69.29 | 66.43 | 67.14 | 70.00 | 70.36 | 64.29 | 69.29 | 68.10 | **74.64** |
| 233094 | 83.77 | 83.17 | 84.36 | **84.39** | 83.36 | 80.82 | 83.17 | 83.20 | 81.98 | 83.77 | 81.47 | 78.65 | 83.20 | 82.60 | 82.58 |
| 233096 | 70.27 | 66.56 | 71.95 | 76.43 | 75.44 | 77.40 | 71.95 | 65.31 | **78.31** | 72.43 | 76.84 | 74.94 | 67.33 | 72.98 | 74.38 |
| 233099 | 76.89 | 77.92 | 75.95 | 78.23 | 76.38 | 78.38 | 76.75 | 75.56 | 78.61 | 78.67 | **82.56** | 82.23 | 76.99 | 78.52 | 82.44 |
| 233102 | 61.00 | 62.89 | 61.32 | 63.57 | 56.67 | 60.79 | 59.99 | 43.04 | 60.77 | 61.95 | 63.30 | 63.49 | **64.03** | 63.75 | 63.92 |
| 233103 | 87.51 | 87.02 | 88.25 | 87.03 | 87.22 | 85.90 | 87.99 | 87.64 | 85.90 | 87.12 | 87.26 | 86.59 | 86.74 | **88.39** | 86.62 |
| 233104 | **99.97** | 99.96 | 99.96 | 99.94 | 2.57 | **99.97** | 99.95 | 92.77 | **99.97** | 99.95 | 99.96 | **99.97** | 99.96 | 99.96 | 99.95 |
| 233106 | 62.83 | **68.90** | 62.46 | 65.70 | 62.16 | 61.85 | 61.89 | 44.63 | 62.05 | 66.29 | 65.43 | 64.99 | 66.50 | 67.04 | 68.11 |
| 233107 | 95.92 | 95.93 | 96.01 | 96.36 | 95.23 | 95.76 | 95.77 | 95.37 | 96.22 | 96.52 | 96.10 | 96.55 | 95.98 | 96.23 | **96.83** |
| 233108 | 87.50 | 91.20 | 85.65 | 87.96 | 50.00 | 93.98 | 92.59 | 94.91 | 94.44 | 94.44 | 93.06 | 95.37 | 91.67 | 90.74 | **95.83** |
| 233109 | 67.84 | **73.68** | 66.52 | 68.20 | 66.45 | 65.20 | 66.89 | 66.74 | 67.03 | 68.64 | 67.32 | 70.18 | 66.23 | 68.42 | 67.62 |
| 233110 | 78.08 | 72.58 | 72.70 | 83.40 | 66.93 | 72.74 | 74.12 | 70.16 | 74.76 | 74.09 | 85.71 | 85.76 | 72.34 | 83.14 | **85.99** |
| 233112 | 73.63 | 74.68 | 73.37 | 74.33 | 77.36 | 73.86 | 72.91 | 72.06 | 74.35 | 72.08 | 76.23 | 75.74 | 72.48 | 76.35 | **80.07** |
| 233113 | 99.47 | 99.89 | 99.92 | 99.87 | 55.86 | 98.11 | 99.46 | 90.60 | 98.11 | 99.94 | 99.92 | 99.91 | 99.88 | 99.89 | **99.95** |
| 233114 | 67.75 | 68.90 | 68.81 | 69.11 | 67.36 | 68.08 | 67.44 | 67.70 | 68.56 | 68.59 | 71.93 | 73.13 | 67.80 | 66.87 | **73.55** |
| 233115 | 86.27 | 85.79 | 88.73 | 86.44 | 87.26 | 87.74 | 88.39 | 87.74 | 88.39 | 88.73 | 88.25 | **88.90** | 87.91 | 86.27 | 87.09 |
| 233116 | 97.44 | **100.00** | 96.79 | 97.44 | 87.35 | 99.47 | 99.44 | 97.46 | 99.69 | 99.37 | 97.64 | 99.04 | 97.44 | 99.69 | 99.59 |
| 233117 | **94.81** | 92.86 | 93.51 | 93.51 | 90.48 | 93.72 | 92.86 | 92.64 | 93.72 | 93.51 | 93.07 | 93.72 | 93.94 | 94.59 | 93.72 |
| 233118 | 90.46 | 90.86 | 90.73 | 90.75 | 81.72 | 89.91 | 90.69 | 86.69 | 90.06 | 91.11 | 91.09 | 91.88 | 90.70 | 90.51 | **91.95** |
| 233119 | 97.06 | 93.76 | 97.79 | 96.08 | 92.15 | 97.83 | 97.16 | 87.08 | **98.14** | 96.50 | 96.50 | 97.51 | 97.33 | 97.24 | 97.47 |
| 233120 | 50.26 | 50.95 | 51.29 | 50.50 | 51.63 | 50.92 | 50.17 | 50.23 | 51.00 | 50.72 | 52.35 | 52.10 | 50.41 | 50.30 | **52.67** |
| 233121 | 96.12 | 97.83 | 96.45 | 96.74 | 92.40 | 95.31 | 96.34 | 91.38 | 95.15 | 97.52 | 97.88 | 97.80 | 96.88 | 97.00 | **98.37** |
| 233122 | 16.84 | 22.26 | 17.20 | 19.65 | 20.90 | 24.53 | 16.77 | 18.71 | 24.35 | 23.62 | 23.43 | 24.10 | 17.52 | 23.98 | **27.70** |
| 233123 | 51.51 | 51.74 | 50.86 | 53.16 | 56.11 | 53.58 | 49.65 | 49.88 | 51.94 | 51.22 | 60.98 | 61.67 | 51.13 | 55.12 | **65.29** |
| 233124 | 65.08 | 66.82 | 65.57 | 66.56 | 66.15 | 57.71 | 65.26 | 64.97 | 58.04 | 67.24 | 70.03 | 68.84 | 66.86 | 67.00 | **71.67** |
| 233126 | 90.64 | 58.17 | 90.42 | 92.94 | 92.60 | 93.99 | 90.45 | 88.55 | 93.98 | 93.58 | 93.86 | 93.87 | 92.97 | **94.10** | 94.02 |
| 233130 | 87.76 | 87.81 | 88.98 | 88.99 | 70.72 | 87.99 | 50.00 | 85.25 | 88.35 | 92.43 | 50.00 | **95.81** | 94.92 | 91.19 | 92.53 |
| 233131 | 70.94 | 69.28 | 71.59 | 70.94 | 71.31 | 72.14 | 71.59 | 71.59 | 72.32 | 70.94 | 72.69 | 72.42 | 70.76 | 70.76 | **74.26** |
| 233132 | 96.93 | 98.62 | 97.52 | 97.14 | 94.58 | 96.85 | 97.00 | 97.27 | 96.90 | 98.66 | 98.14 | **99.15** | 96.81 | 96.73 | 99.05 |
| 233133 | 63.71 | 65.11 | 65.00 | 66.05 | 64.57 | 66.21 | 62.82 | 64.33 | 65.98 | 68.75 | 66.58 | 66.28 | 64.36 | 64.81 | **69.18** |
| 233134 | 78.05 | 75.87 | 79.05 | 78.22 | **80.38** | 78.38 | 76.88 | 78.38 | 78.38 | 76.88 | 77.38 | 76.54 | 76.88 | 76.21 | 79.22 |
| 233135 | 93.07 | 92.49 | 92.10 | 93.17 | 93.17 | 92.10 | 93.17 | 93.27 | 92.10 | 92.58 | 92.68 | **94.53** | 93.75 | 93.36 | 94.05 |
| 233137 | 91.91 | 93.71 | 92.16 | 92.56 | 90.38 | 91.58 | 91.36 | 88.09 | 91.60 | 92.72 | 92.48 | 92.39 | 92.95 | 92.72 | **94.01** |
| 233142 | 92.33 | 96.70 | 92.90 | 92.35 | 63.59 | 95.47 | 91.43 | 93.60 | 95.56 | 93.47 | 93.81 | 93.25 | 92.60 | 93.85 | **97.17** |
| 233143 | 50.00 | 92.30 | **92.76** | 50.00 | 70.81 | 90.28 | 50.00 | 50.31 | 89.26 | 50.00 | 50.00 | 50.00 | 92.26 | 50.00 | 92.53 |
| 233146 | 63.12 | 60.06 | 62.79 | 64.16 | 63.39 | 64.42 | 63.52 | 54.64 | 64.21 | 64.26 | 64.05 | **64.57** | 64.41 | 64.37 | 64.28 |

Table 11: **Detailed Table of Results.** The test set performance for the plain network, individual regularization methods and for the regularization cocktails.

| Task Id | Fixed LR Cocktail | Dynamic LR Cocktail |
|---|---|---|
| 233090 | 89.270 | **90.000** |
| 233091 | **99.850** | **99.850** |
| 233092 | **61.461** | 56.518 |
| 233093 | 98.000 | **98.250** |
| 233088 | **74.643** | 64.881 |
| 233094 | **82.576** | 79.654 |
| 233096 | **74.381** | 70.058 |
| 233099 | 82.443 | **82.551** |
| 233102 | **63.923** | 63.884 |
| 233103 | 86.619 | **87.854** |
| 233104 | 99.953 | **99.967** |
| 233106 | 68.107 | **69.081** |
| 233107 | **96.826** | 96.446 |
| 233108 | **95.833** | 95.370 |
| 233109 | 67.617 | **67.836** |
| 233110 | 85.993 | **86.596** |
| 233112 | **80.073** | 78.985 |
| 233113 | **99.948** | 83.263 |
| 233114 | **73.546** | 73.276 |
| 233115 | 87.088 | **88.077** |
| 233116 | 99.587 | **99.690** |
| 233117 | 93.723 | **93.939** |
| 233118 | 91.950 | **91.964** |
| 233119 | 97.471 | **98.039** |
| 233120 | **52.668** | 52.204 |
| 233121 | 98.370 | **98.522** |
| 233122 | 27.701 | **28.008** |
| 233123 | **65.287** | 63.293 |
| 233124 | 71.667 | **72.243** |
| 233126 | **94.015** | 93.930 |
| 233130 | 92.535 | **94.894** |
| 233131 | **74.262** | 72.140 |
| 233132 | 99.049 | **99.404** |
| 233133 | **69.183** | 68.877 |
| 233134 | **79.217** | 78.887 |
| 233135 | 94.045 | **94.435** |
| 233137 | **94.010** | 93.961 |
| 233142 | **97.175** | 97.106 |
| 233143 | 92.531 | **92.592** |
| 233146 | 64.280 | **64.362** |

Table 12: The test set performances of the regularization cocktails with a fixed initial learning rate value and a dynamic learning rate chosen by BOHB.

| Task Id | MLP | MLP + D | MLP + S | XGB. + ES | XGB. + ES + ENC | XGB. + ENC | CatBoost | TabN. + ES | AutoGL. + HPO | MLP + C |
|---|---|---|---|---|---|---|---|---|---|---|
| 233090 | 84.131 | 86.916 | 87.354 | **90.000** | 89.000 | 89.000 | **90.000** | 66.678 | 78.854 | 89.270 |
| 233091 | 99.701 | **99.850** | 99.687 | 99.701 | 99.687 | 99.701 | 99.197 | 99.687 | 99.238 | **99.850** |
| 233092 | 37.991 | 38.704 | 51.321 | 50.631 | 48.841 | 48.779 | 51.882 | 30.447 | 51.943 | **61.461** |
| 233093 | 97.750 | 98.000 | 97.000 | 96.750 | 98.000 | **98.250** | 97.250 | 97.250 | 97.500 | 98.000 |
| 233088 | 69.405 | 68.095 | 66.429 | 66.786 | 65.595 | 68.214 | 67.976 | 63.571 | **77.738** | 74.643 |
| 233094 | 83.766 | 82.603 | **85.444** | 75.081 | 73.241 | 74.405 | 79.732 | 66.061 | 82.624 | 82.576 |
| 233096 | 70.274 | 72.980 | **79.429** | 50.000 | 62.713 | 65.027 | 65.660 | 60.065 | 72.219 | 74.381 |
| 233099 | 76.893 | 78.520 | 81.354 | 78.790 | 78.917 | 79.529 | 81.045 | 76.715 | **82.768** | 82.443 |
| 233102 | 60.997 | 63.754 | 62.709 | 60.312 | 61.734 | 60.703 | 61.001 | 54.327 | 61.119 | **63.923** |
| 233103 | 87.514 | **88.387** | 87.886 | 87.079 | 87.841 | 87.553 | 87.579 | 79.027 | 87.975 | 86.619 |
| 233104 | 99.971 | 99.962 | 99.962 | 99.962 | **99.977** | 99.967 | 99.968 | 99.958 | 99.971 | 99.953 |
| 233106 | 62.831 | 67.035 | 63.560 | 98.668 | **98.850** | 98.721 | 69.889 | 53.774 | 56.662 | 68.107 |
| 233107 | 95.917 | 96.232 | 96.273 | 96.460 | 96.785 | **97.263** | 97.049 | 96.232 | 96.212 | 96.826 |
| 233108 | 87.500 | 90.741 | 93.519 | 95.370 | 93.519 | 93.519 | 95.370 | 85.648 | 94.907 | **95.833** |
| 233109 | 67.836 | 68.421 | **69.956** | 57.237 | 57.456 | 62.427 | 59.576 | 64.547 | 69.152 | 67.617 |
| 233110 | 78.076 | 83.145 | **86.137** | 73.329 | 72.087 | 72.252 | 74.152 | 67.893 | 83.097 | 85.993 |
| 233112 | 73.627 | 76.345 | 75.809 | 72.849 | 72.875 | 73.730 | 71.578 | 60.440 | 76.312 | **80.073** |
| 233113 | 99.475 | 99.892 | 99.494 | 97.143 | 98.571 | 98.571 | 98.571 | 63.273 | 83.851 | **99.948** |
| 233114 | 67.752 | 66.873 | 67.985 | 72.781 | 72.779 | 72.800 | 72.809 | 72.431 | 73.167 | **73.546** |
| 233115 | 86.268 | 86.268 | 83.159 | **91.186** | 50.000 | 89.376 | 91.016 | 87.428 | 86.300 | 87.088 |
| 233116 | 97.442 | 99.690 | 97.757 | 96.085 | 93.512 | 97.870 | **100.000** | 88.195 | 97.565 | 99.587 |
| 233117 | **94.805** | 94.589 | 93.723 | 93.723 | 91.991 | 93.290 | 92.424 | 91.126 | 91.126 | 93.723 |
| 233118 | 90.464 | 90.507 | 89.007 | 91.064 | **91.136** | 91.093 | 90.893 | 89.171 | 90.964 | 91.950 |
| 233119 | 97.061 | 97.237 | 90.997 | 90.938 | 90.085 | 90.748 | 82.150 | 88.403 | **99.270** | 97.471 |
| 233120 | 50.262 | 50.301 | 51.917 | 51.708 | 52.257 | 52.190 | 52.083 | 51.340 | 52.452 | **52.668** |
| 233121 | 96.125 | 97.000 | 2.174 | 91.967 | 93.082 | 92.533 | 95.777 | NaN | 97.299 | **98.370** |
| 233122 | 16.836 | 23.983 | 18.618 | 19.004 | 22.969 | 23.190 | 23.213 | 20.945 | 26.466 | **27.701** |
| 233123 | 51.505 | 55.118 | 56.543 | 54.858 | 54.705 | 55.295 | 55.453 | 55.212 | 60.209 | **65.287** |
| 233124 | 65.081 | 66.996 | 63.064 | 61.342 | 63.472 | 64.717 | 62.660 | 63.458 | 67.986 | **71.667** |
| 233126 | 90.639 | 94.099 | 93.701 | 93.880 | 94.055 | 93.979 | 93.932 | 49.790 | **95.172** | 94.015 |
| 233130 | 87.759 | 91.194 | **93.720** | 89.546 | 87.304 | 86.607 | 87.910 | 90.012 | 91.776 | 92.535 |
| 233131 | 70.941 | 70.756 | 69.926 | 74.815 | 74.170 | **75.738** | 73.708 | 69.649 | 73.801 | 74.262 |
| 233132 | 96.930 | 96.733 | 96.462 | 97.107 | 96.467 | 96.519 | **99.259** | 97.266 | 98.856 | 99.049 |
| 233133 | 63.707 | 64.814 | 64.877 | 68.952 | 68.911 | 70.331 | **71.708** | 63.811 | 66.074 | 69.183 |
| 233134 | 78.048 | 76.211 | 75.539 | 77.867 | 77.197 | 78.370 | 80.052 | 75.527 | **80.211** | 79.217 |
| 233135 | 93.070 | 93.363 | **95.217** | 95.119 | 94.827 | 94.924 | 95.119 | 92.681 | 93.364 | 94.045 |
| 233137 | 91.905 | 92.724 | 91.513 | 89.141 | 90.793 | 89.026 | NaN | 91.334 | 93.551 | **94.010** |
| 233142 | 92.331 | 93.852 | 92.707 | 95.019 | 93.870 | 94.047 | 85.379 | 89.384 | 96.207 | **97.175** |
| 233143 | 50.000 | 50.000 | 50.000 | 88.263 | 87.749 | 89.791 | 92.345 | 85.701 | **92.829** | 92.531 |
| 233146 | 63.125 | **64.367** | 50.000 | 57.866 | 58.143 | 58.324 | 56.563 | 50.340 | 59.203 | 64.280 |

Table 13: The results for the remaining baselines used in our experiments. Each performance represents the test accuracy of the incumbent configuration after being refit.