# OpenReview forum: "Well-tuned Simple Nets Excel on Tabular Datasets"
_NeurIPS.cc/2021/Conference — NeurIPS 2021 Poster_

### Official Review · Reviewer_BhRe · 2021-07-13

**Rating:** 7
**Confidence:** 4

**Summary:**

On a wide selection of tabular data datasets, this paper showed empirically that well regularized MLPs outperform recent tabular data specific architectures as well as classical ML methods on tabular data classification tasks. In particular, an array of regularizes was studied experimentally, and a principled choice of methods and hyperparameters leads to  high performing MLPs.

**Ethical Concerns:**

I have no ethical concerns.

**Limitations And Societal Impact:**

Yes, both the "Limitations" and "Societal Impact" are adequate.

**Main Review:**

This paper makes a compelling case that MLPs have potential to outperform domain specific architectures. The results and figures presented show careful assessment of the experimental design. While there is not much motivation provided, the fact that MLPs  are capable, albeit with some help, to beat older methods is maybe not surprising, but the fact that a well-regularized MLP can outperform TabNet and NODE is interesting. It begs the question: Could the TabNet and NODE be regularized, or otherwise have their hyperparameters tuned and perform better yet?

Strengths: Thorough and principled experimental demonstration of the main point of the paper. Informative and legible plots, in particular, Figure 1 is easy to digest. Clear and concise writing style.

Weaknesses: There is little mentioned about why a given "regularization cocktail" might be a good combination. Also, the comparison to recent neural architectures is limited in that the recent methods are not as optimized as the MLPs. To the authors' credit, this limitation is noted in the final section of the paper and it may not harm the significance of these findings.

In summary, the main finding is important for the community to understand, even if it is not wildly surprising. Intuition and explanation behind the best performing combinations of regularizers would be nice.

___
Reaction to authors' response: I went back to reread existing work on NNs for tabular data. It was correctly pointed out by the authors that even works like TabNet (Arik and Pfister 2020) and TabTransformer (Huang et al. 2020) do not make an entirely compelling case that NNs outperform XGBoost. For that reason, I think additional empirical works like this paper are valuable to the community.

**Time Spent Reviewing:**

5

---

> ### Author Response · Authors · 2021-08-10
> **Response to Reviewer BhRe**
>
> Thanks a lot for your positive criticism and the valid arguments.
>
> **On cocktails of other neural architectures:** You have a valid point here. Although we tuned the hyperparameters for NODE and TabNet, we agree that in principle these methods can be regularized further with a cocktail of regularizers, and their results might improve. However, please note that our research hypothesis is not that simple MLPs are the best possible neural architecture, or that they are the only candidate architecture for being well-regularized with the proposed cocktail. Instead, we raised the hypothesis that “even” simple MLPs can be very competitive if well-regularized, to the point of actually outperforming the state-of-the-art models on tabular data. We will clarify this further in the limitations section.
>
> **On the surprising findings:** We understand the reviewer’s comment that the results feel unsurprising. It seems the belief of a deep learning enthusiast would be that “of course advanced neural networks outperform gradient-boosted trees”. However, the bitter truth was the opposite, as previous studies (references 38, 22 in our manuscript) showed that gradient-boosted trees (concretely XGBoost) actually outperformed neural networks on tabular data. Therefore, the impression by some in the community that “deep learning outperforms XGBoost” was empirically unsupported. In this work, for the first time we demonstrate the superiority of neural networks over gradient-boosted trees in terms of prediction accuracy on tabular data.
>
> Thanks again, and please feel free to raise any further point that was either left unanswered, or is a follow-up of our clarifications.

---

### Official Review · Reviewer_9cio · 2021-07-17

**Rating:** 6
**Confidence:** 3

**Summary:**

This paper aims to study how regularizing basic MLP models can make them as effective as the best available models for tabular data (GBDT and more sophisticated/recent NN architectures like TabNet/NODE). The main contribution is to propose to search among 13 regularizers to find a combination that maximizes validation score, where the search is done via standard hyperparameter optimization techniques, and they find the resulting regularized MLPs outperform GBDT and more recent NN architectures across many tabular datasets.

**Ethical Concerns:**

I don't have remaining omitted ethical concerns regarding this paper.

**Limitations And Societal Impact:**

I believe the authors have adequately addressed societal impact & limitations of their work.
One unaddressed limitation is that given limited training time, it's unclear whether training both MLP+D and XGBoost followed by model-selection would be competitive to MLP+C (since XGBoost is presumably much quicker to train and both MLP+D and XGBoost presumably require less HPO trials than MLP+C). Also the paper didn't consider more advanced boosting variants like CatBoost and used questionable label-encoding of categoricals for XGBoost.


**Main Review:**

Through a careful empirical study, this paper provides a number of important results with practical significance. Namely that MLPs can outperform XGB across a wide dataset range and that
In particular, the conclusion that "well-regularized simple deep MLPs outperform specialized neural architectures" is an important one for the community to be informed of. However the main weakness of this paper in my eyes is it does not actually answer the question of how I should use MLPs for tabular data to get the best performance. This is because the authors did not consider how to split a limited HPO budget into tuning regularizers vs other hyperparameters like the network architecture, learning rates, etc. I also still have questions regarding the details of baselines in the empirical  comparisons.


- What makes these "regularization cocktails" different than just a hyperparameter-tuned MLP?
It seems the authors are considering implicit bias aspects of the NN as "regularization" as well, in which case wouldn't also the activation function and other hyperparameters that affect implicit bias also fit this definition of regularization?
In fact the authors say their fixed hyperparameters "have been tuned for maximizing the performance of an unregularized neural network on our dataset collection"; more details about this process would be nice to see.
Why not tune hyperparameters like number of layers, learning rate, hidden layer size, etc? Is there some practical advantage to tuning only the regularization cocktail in MLP+C vs other hyperparameters? Given a limited HPO budget, it's still not clear to me how I should design a MLP search space and whether I should only search over regularizers or also the hyperparameters I just mentioned.


- Tuning num_round as a hyperparameter of XGBoost is a bit odd. Normally one just sets this large and  relies on early-stopping. Tuning num_round might lead to the effective number of meaningful XGB HPO trials completed within the budget much smaller than for other models. The authors state they also tried XGB + ES, but did not clarify precisely what was done there. In particular did you remove num_round from the search space and if so to what value was it set?


- The authors are missing reference to SeLU paper:

Self-Normalizing Neural Networks
Klambauer et al. (2017)
https://arxiv.org/abs/1706.02515

And an empirical comparison against this technique, which seemed to also perform well on tabular data in the original paper.


- More details of MLP+D baseline would be good to have. Eg. Was the DO-active hyperparameter fixed = True and the same number of HPO trials / total compute run for this baseline?


- Is multifidelity HPO method like BOHB able to accurately assess effect of something like Dropout? Wouldn't dropout appear to be harmful in the early stages of MLP training since it slows down the optimization (although allowing the optimization to eventually reach a better-generalizing solution)?


- Table 1 caption should include references for each regularization method. Some are not fully clear in their details such as "Multi-branch choice".


- If the authors claim "MLP with Regularization is all you Need", then why only compare against the NN model in AutoGluon Tabular, why not compare against the whole automl system which as far as I know combines NN with GBDT models?

- The authors compared against AutoGluon without HPO and with only NN models, claiming its stacking is superior to HPO. However to me, it seems stacking is not nearly as useful with only NN models and is intended for settings where many different models are trained. It would be good to see AutoGluon + HPO as well if only its NN models are being used.


- It would be good to compare plot the performance vs number of HPO trials on a dataset where MLP+C is better than XGB for both methods. Is it the case the XGB performance plateaus after certain number of HPO trials, while MLP+C performance continues to increase with additional HPO?  If so, this could indicate the XGB hyperparameter search space is too small relative to the allotted compute/runtime.
Relatedly it would be good to know how sensitive the results are to the specific compute/runtime limits chosen for on all methods here.


- Were any of the regularizers never selected in the cocktail? These could be helpful to point out.


- It would be interesting if the authors could investigate what characteristics of a dataset make it such that MLP+C >>> XGB versus the reverse. For example XGB crushes MLP on the ldpa dataset, and is crushed by MLP+C on arrhythmia.


- It may also be interesting to produce a figure like sklearn's classifier comparison:
https://scikit-learn.org/stable/auto_examples/classification/plot_classifier_comparison.html
showing the effects of each regularizer on the MLP decision boundary on a toy 2d dataset.


### Update after reading author response to my original review ###
I find the paper stronger after the additional experiments the authors have run and have slightly increased my score. That said, I still find this a borderline paper. The biggest remaining concerns I have that do not seem to have been addressed in the authors' response include:

- the title needs to be changed in my opinion. It's clearly not close to "all you need" if the authors acknowledge autogluon with all of its different models will be stronger than just their regularized-MLP.

- the authors should more clearly explain what criteria they use to decide whether a hyperparameter is regularization-related or not. Again, to me almost every hyperparameter affects inductive bias in some way. There should be a more systematic explanation for why these particular hyperparameters were considered in the search space but not other standard hyperparameters that are typically tuned in MLPs.

- the early-stopping experiment of XGBoost with num_round = 4000 could still be made more convincing. In particular, num_round could be set to a much greater value no, such that early-stopping is guaranteed to hit?  I feel at the very least num_round should be set to the maximal value such that XGBoost training runtimes do not exceed the MLP training runtimes.

- autogluon with just its MLP model should be run with HPO. I believe the autogluon recommendation of "use stacking instead of HPO" only applies when the full system with all of its models are being utilized. If you are only utilizing MLP models, then I believe HPO would outperform stacking.  To reiterate: stacking seems a bit odd to me when the ensemble only contains one type of model as in your usage.

That said, I believe the authors can easily address these in a revision, and I do think this is a high-impact paper with an important conclusion for the ML community. Papers like this are difficult to write and sometimes harshly judged since there are uncountably many comparisons/analyses that could've been run, but I think the choices made by the authors appear reasonable to me and sufficient to convincingly present some important conclusions.

**Time Spent Reviewing:**

4

---

> ### Author Response · Authors · 2021-08-10
> **Response to Reviewer 9cio - Part 1**
>
> **On designing the search space:** Thanks for raising the important argument on the implicit regularization hyperparameters. We agree that the hyperparameters governing the choice of the architecture, the activation functions, the capacity of the model (num weights), as well as the optimizer settings are also implicit regularization mechanisms. However, we did not expand the HPO design space (i) to avoid criticism/skepticism by practitioners/reviewers that there are too many hyperparameters (notice there are dozens of potential implicit design hyperparameters for neural networks), and (ii) because the results were already statistically significant against all the baselines with the actual compact HPO design space. We do not see this as a weakness, because in practice expanding the HPO design space with more hyperparameters will very likely improve empirical results even further (at the expense of needing more HPO runtime), which would only reinforce our key message that MLPs with a properly tuned mix of regularizations can statistically significantly outperform GBDTs. We believe that our result opens the door to many follow-up works improving our results in many different ways (e.g., also optimizing the different components pointed out by the reviewer, different architectures, choosing between different optimizers, weight initialization strategies; making this optimization much more efficient via meta-learning; in-depth studies to understand when which type of regularization is most effective; etc).
>
> **On SeLU:** We implemented and ran the baseline of MLP with SeLU (MLP + S) activation on the exact same protocol as the other baselines. The results are shown in the plot below:
>
> https://github.com/ConferencePaperSubmission/RegularizationCocktail/blob/main/info_plots/selu_cd_diagram.pdf
>
> As can be seen from the results, MLPs with SeLU activations indeed perform better than MLPs with ReLU activations (as the reviewer expected), even though the difference is not statistically significant among the two variants, with a Wilcoxon signed rank test having a p-value of 0.15. Still, we fully agree that we should include this option in the regularization choices. Thanks for the suggestion!
>
>
> **On other variants of Gradient-Boosted Decision Trees (GBDT):** We agree that it would be useful to also compare to more modern versions of GBDTs in order to make the comparison as fair and comprehensive as possible. We therefore now also ran CatBoost (official implementation) in our empirical protocol. We used the exact hyperparameter space detailed in the CatBoost paper (https://arxiv.org/pdf/1706.09516.pdf) and gave CatBoost the same HPO budget as all other methods. Please see a comparison of CatBoost vs. our method and the other baselines in the plot below:
>
> CatBoost vs GBDT implementations and the Regularization Cocktails:
> https://github.com/ConferencePaperSubmission/RegularizationCocktail/blob/main/info_plots/catboost_gbdt_cd_diagram.pdf
>
> CatBoost vs all the baselines and the Regularization Cocktails:
> https://github.com/ConferencePaperSubmission/RegularizationCocktail/blob/main/info_plots/catboost_baselines_cd_diagram.pdf
>
>
> We note that so far the Catboost results for this comparison have only been partially completed and the diagrams provided above are built based on only 37 out of 40 datasets. We will post a link with the full results once all jobs are completed. The current results strongly indicate that our regularization cocktails are superior to CatBoost with a statistically significant margin, which further supports the state-of-the-art status of our proposed method. Please note that we now demonstrate that well-regularized MLPs significantly outperform **three** different implementations of GBDT (CatBoost, XGBoost and Autosklearn/scikit-learn). This is in fact the first paper to show that MLPs can outperform GBDT in a fair (thorough HPO for all methods) and large-scale empirical protocol with proper statistical significance measures. If the reviewer has any other suggestions on another GBDT implementation to try, or another hyperparameter space to try, we will gladly run those experiments, too.
>
> **Update 27 August 2021**
>
> The plots related to the CatBoost baseline have now been updated with the extended results from 39 out of 40 datasets. Unfortunately, for one dataset CatBoost did not manage to finish any configurations in time (the dataset features the most examples in our benchmark.)
>
> **On the dropout baseline:** The reviewer's understanding is correct, the dropout method was the only activated regularization technique as explained in Section 6. We will clarify this in the paper, too, to avoid confusion.
>
> **On multi-fidelity HPO and regularization:** We agree, your intuition is correct and matches perfectly what we observed in our experiments. Hyperparameter configurations with a strong regularization converge later than the configurations with weak regularization, leading to a poor rank correlation between a) the validation loss of configurations at a low-budget/fidelity and b) the respective validation loss at the full budget. In such cases, BOHB (or even HyperBand) would be at risk of discarding very good configurations (strong regularization) at low budgets. One could likely carefully tweak the budgets and Hyperband’s eta parameter to make the rank correlations large enough, but to avoid such tweaks in this work we simply ran BOHB with only the largest budget (such that all configurations have converged) and gave ample HPO time to ensure that a large number of configurations are considered. This  strategy was sufficient for the validity of our experiments, but we believe this indeed raises a big question for the AutoML research community in designing future multi-fidelity HPO methods which better capture the rank correlation among budgets.
>
> **On early stopping for XGBoost:**
> There are indeed two possibilities for setting XGBoost’s num_rounds hyperparameter: (1) tuning it on a per-dataset basis or (2) setting it to a high value and using early stopping. In order to be maximally fair to XGBoost, we tried both of these methods. In the version with early stopping, we set the number of rounds to a large value of 4000. We found this version to not perform better; we ablate this change in the figure below and will add this figure & information to the camera ready version.
>
> https://github.com/ConferencePaperSubmission/RegularizationCocktail/blob/main/info_plots/xgboost_early_stopping_ablation.pdf
>
> As can be seen by the critical difference diagram displayed above, the XGBoost variant that does not make use of early stopping performs better than the version that uses early stopping, and the difference is statistically significant.
>
> **On the use of label encoding in  XGBoost:** Categorical values can be either encoded using a one-hot encoding or using a label encoding. While in our paper we report results with label encoding, to be fair, we also ran an additional experiment where BOHB could choose the best encoding among the two options for each dataset. The plot below shows that this hyper-parameterized choice of the encoding does not improve the performance compared to only using label encoding. The XGBoost runs with the categorical hyperparameter encoding (choice between one-hot enc. vs label enc.) are denoted with the abbreviation + ENC in the cd-diagram.
>
> https://github.com/ConferencePaperSubmission/RegularizationCocktail/blob/main/info_plots/xgboost_conditional_encoding_ablation.pdf
>
> From the diagram above it can be seen that both versions of XGBoost with/without early stopping perform worse when the encoding choice is a hyperparameter, and that this difference in results is statistically significant.
> The results above are still preliminary since the experiments for larger datasets are still running. Specifically, for the XGBoost with early stopping where the encoding choice was a hyperparameter the diagrams above were built based on the results of 34 out of 40 datasets, while for the non-early stopping version with the same traits the results are based on 23 out of 40 datasets.
>
> **Update 27 August 2021**
>
> The plots related to the XGBoost ablations have now been updated with the full results on all datasets.
>
> **On the diversity of the HPO search space for XGBoost:** Your concern that the search space of XGBoost might be restrictive and the HPO performance plateaus after a few trials is reasonable. As a result, we analysed the number of trials that our method and XGBoost needed until finding the best performing hyperparameter configurations. The distribution of the “num. trials until the best” across the 40 datasets are shown in the plot below:
>
> https://github.com/ConferencePaperSubmission/RegularizationCocktail/blob/main/info_plots/comparison_number_trials.pdf
>
> As can be seen, XGBoost actually needed more trials than our cocktails to find the best performing configuration, which speaks against a more restrictive HPO search space for XGBoost.
> We also again refer to the results shown above (reply to reviewer V9FE), which show that our method already statistically significantly outperformed XGBoost after as little as 1 hour.

---

> > ### Author Response · Authors · 2021-08-27
> > **Response to Reviewer 9cio**
> >
> > **On the title change:** We agree with the reviewer regarding potential misinterpretations of the “All you need” term. Therefore, we propose the following title change: “Well-tuned Simple Nets Excel on Tabular Datasets”.
> >
> > **On the choice of hyperparameters related to regularization:** To avoid becoming repetitive, we invite the reviewer to read our response to reviewer VweU on the matter.
> >
> > **On the choice of the num_round hyperparameter for XGBoost:** To address the reviewer’s concern, we provide the following figure that shows the distribution of the number of XGBoost rounds until the early stopping heuristic was triggered across 40 datasets:
> > https://github.com/ConferencePaperSubmission/RegularizationCocktail/blob/main/info_plots/nr_rounds_xgboost.pdf
> >
> > As can be seen from the plot, XGBoost always early stops before reaching the maximum limit of 4000 rounds (in fact XGBoost needs at most ca. 1000 rounds in our datasets). Increasing the limit beyond 4000 would play no role in the behavior of XGBoost with early stopping.
> >
> > **On running AutoGluon with HPO instead of stacking:** We understand the reviewer’s concern that HPO might be competitive against stacking, given that all base models of the stacking ensemble are neural networks and lack diversity (although each base network is trained on a bagging mode, based on data subsets). As a result, following the reviewer’s suggestion, we ran AutoGluon with hyperparameter optimization instead of stacking, using the library’s default search space found at:
> > https://github.com/awslabs/autogluon/blob/62cb6ba217a7ded180e6b66cb86d5a7f4321ddd3/tabular/src/autogluon/tabular/models/tabular_nn/hyperparameters/searchspaces.py
> >
> > The empirical performance of AutoGluon with stacking vs. with HPO and the other baselines is presented in the following critical difference diagrams:
> > https://github.com/ConferencePaperSubmission/RegularizationCocktail/blob/main/info_plots/autogluon_hpo_comparison_baselines.pdf
> >
> > The figure shows that AutoGluon that features HPO has a better rank compared to AutoGluon that features stacking, which empirically validates the reviewer's intuition on the benefit of HPO over stacking. However, the difference between the results of AutoGluon that features stacking and AutoGluon that features HPO is not statistically significant. In contrast, our regularization cocktails manage to outperform both AutoGluon-Tabular variants significantly.

---

> > > ### Comment · Reviewer_9cio · 2021-08-27
> > > **Thanks for the new results**
> > >
> > > I appreciate the authors' extra experiments, and no longer have any big concerns about this paper.

---

> ### Author Response · Authors · 2021-08-10
> **Response to Reviewer 9cio - Part 2**
>
> **On the comparison to AutoGloun:** The “all you need” is obviously just a playful slogan borrowed from prior examples. Please note that our contribution here is in terms of prediction models, concretely showing that well-regularized MLPs can outperform GBDT in the realm of tabular datasets. As also argued in the reply to Reviewer V9FE above, AutoGluon-Tabular is a comprehensive AutoML system employing many model families, and here we rather seek to compare one model family (neural nets) against another one (GBDT).
>
> Your further comments on the diversity of neural networks in the pure Deep Learning variant of AutoGloun, as well as on stacking vs. HPO, are very interesting. The diversity of neural networks in that library comes from i) bagging as every model is trained on subsets of the training data, and ii) different feature spaces because each neural network receives as features the predictions from the diverse models of one layer below plus the original dataset features. In that sense, every neural model in AutoGluon’s bagging+stacking approach is diverse in the ensembling notion of the correlation between prediction errors. Regarding the other point on AutoGluon’s HPO we emphasize that the authors of the work do not use HPO (see https://arxiv.org/pdf/2003.06505.pdf). Specifically, we provide the following information from the official documentation of AutoGluon (https://auto.gluon.ai/api/autogluon.task.html): **‘As an example, to get the most accurate overall predictor (regardless of its efficiency), set `presets='best_quality'`’** which in turn, does not feature HPO.
> Additionally, **“It is currently not recommended to use hyperparameter_tune with auto_stack due to potential overfitting”**. Their rationale is that bagging and stacking is so good at reducing variance that it achieves strong results out of the box. Out of all the deep learning approaches we evaluated, Autogluon was actually clearly the strongest competitor, and not significantly worse than a well-tuned XGBoost.
>
> **On whether any regularizer is never selected:** Please find the histogram of the selected ingredients of the regularization cocktails in Figure 5 of the supplementary material.
>
> We hope that we could clarify your concerns and we would very much appreciate it if you considered increasing your score. Please feel free to raise any further concerns or questions and we will be happy to provide our clarifications.

---

### Official Review · Reviewer_VweU · 2021-07-17

**Rating:** 7
**Confidence:** 3

**Summary:**

This paper proposes that properly optimized MLPs can perform just as well if not better on tabular data. This is an important topic as it's well know that MLPs have not achieved similar performance as GBDT on data with categorical variables. The paper is well written and reasonably comprehensive in the experiments.

**Ethics Review Area:**

["I don’t know"]

**Main Review:**

This paper proposes that properly optimized MLPs can perform just as well if not better on tabular data. This is an important topic as it's well know that MLPs have not achieved similar performance as GBDT on data with categorical variables. The paper is well written and reasonably comprehensive in the experiments.

Positives:
- This paper is written clearly and have considered a large set of regularization techniques as well as tested on a comprehensive tabular dataset
- The results is surprising and very useful and relevant to the community
- Real world Kaggle datasets are included as well, in addition to the tiny UCI datasets.
- The discussions on limitations is also interesting and useful.

Negatives:
- the optimization being fixed is a negative. Optimizers typically have a big impact on the performance and generalization of MLPs.
- What about the different types of weight initialization for MLPs?

Given that the results are re-generated by the authors, what is not super clear is that are the numbers for non-MLP results consistent with other published papers?

**Time Spent Reviewing:**

2

---

> ### Author Response · Authors · 2021-08-10
> **Response to Reviewer VweU**
>
> We are glad that the reviewer agrees that the results are “surprising and very useful and relevant” to the community, rates the paper as clearly written, with comprehensive experiments and an interesting & useful discussion of limitations.
>
> The reviewers’ two concerns pertain to studying even more degrees of freedom in the deep learning pipeline, namely the choice of optimizer and different types of weight initialization for MLPs. We agree that these are very interesting, but decided to limit the scope of our hyperparameter choices to retain a HPO problem that is computationally feasible, and to still obtain competitive anytime results (please see our reply to Reviewer VNFE above, in which we showed that our regularization cocktails already statistically significantly outperform GBDTs in as little as one hour).
>
> Nevertheless, we fully agree that it is likely possible to obtain even stronger results in future work by also optimizing the choice of optimizer and weight initializer (and indeed also choice of architecture) on a per-dataset basis. We do believe these are some of the many exciting directions our result opens the door to.
>
> Thanks again, and please feel free to raise any further point that was either left unanswered, or is a follow-up of our clarifications.

---

### Official Review · Reviewer_V9FE · 2021-07-24

**Rating:** 6
**Confidence:** 4

**Summary:**

The paper supplies an empirical existence proof that an "appropriately regularized" multi-layer perceptron (MLP)-like model can outperform gradient-boosted decision trees (GBDT) on tabular data. The paper's proposed approach defines a set of parameterized techniques which includes ensembling, architectural changes, and data augmentation, and performs search over combinations from this set. The extensive experimental comparison across 40 tabular datasets shows that MLP-like models can outperform GBDTs and other neural network architectures.

**Limitations And Societal Impact:**

The authors adequately addressed both of these.

**Main Review:**

I appreciated the simplicity of the proposed approach. Showing that Bayesian optimization can select the right combination from such a disparate set of techniques ("soup" came to my mind) is novel, at least to my knowledge. The paper also seems to have addressed some concerns raised in previous submissions with a thorough evaluation performed across a competitor set. However, I have some major concerns on whether the comparison makes sense. The computational cost associated with the proposed approach could be considered impractical wrt/ the standard GBDT solution, but perhaps more work could show this can be reduced. More fine-grained conclusions about why certain techniques, e.g., snapshot ensembles, are necessary and/or sufficient for identifying a high-performing MLP-like model would also have strengthened the submission.

The use of "regularization" to describe the set of techniques over which the proposed method optimizes is misleading. For example, take ensembling: just b/c ensembling has regularizing properties does not make it a regularizer in the traditional sense of a technique to mitigate against overfitting of a single model during training. Conversely, just b/c dropout has ensembling properties does not make it an ensembling technique in the traditional sense of combining the predictions of multiple trained models. Another example is what are termed structural/linearization techniques which include adding skip connects. These really fall under model selection.

A more accurate title would be "Traditional regularization, data augmentation, model selection, model averaging, and deep learning tricks are all you need..."
The value of the contribution comes down to how surprising the community thinks the news that some specific combination of these techniques can yield an MLP-like model or ensemble which, when searched for using 4 days of compute, outperforms GBDT on tabular datasets with <500K rows. If there was evidence that the search could be performed efficiently, that would be hugely beneficial to the message. Unfortunately, there is no finer timing analysis included in the submission.

The overloading of "regularization" in the paper isn't a pedantic issue. It makes fair comparisons wrt/ hypothesis 1 difficult. For example, AutoGluon tabular was run with only neural networks but is designed with general ML models in mind (as this paper notes). This seems artifically limiting if AutoGluon tabular natively, i.e., with AutoGluon's own model and techniques, outperforms GBDT(-like) models; since, if it does, then GBDT should no longer be considered the correct baseline of comparison. More generally, a discussion on why this paper's set of techniques (listed in Table 1) is the "right" set would help in framing which competitors are appropriate vs. unappropriate in a comparison.

* A finer analysis would be helpful to the community. Is there a method rank vs. time plot? method mean+-err balanced accuracy vs. time?

* Are there obvious efficiencies, such as training data-subsampling, in which the computational cost of the proposed approach is retained w/o compromising accuracy?

* How does the absence of one (or multiple) technique(s) from the set of techniques affect performance? E.g., if snapshot ensembles are excluded, how do the results of Table 2 change? Note, this is not a question about which techniques are selected; it is a question about which techniques are generally important/unimportant for the tabular datasets problem.

* It isn't clear how AutoGluon tabular performs natively. If it does not outperform GBDT, then this fact should be included in the paper as it will substantially increase the case for MLP-like solutions.

* Please motivate the selection for the specific set of techniques listed in Table 1.

**Time Spent Reviewing:**

5-7

---

> ### Author Response · Authors · 2021-08-10
> **Response to Reviewer V9FE**
>
> Thanks a lot for your careful review and for raising very valid points (especially on the practical gain vs. GBDT). We will answer your points below and also integrate the respective clarifications and new analytical results in the paper.
>
> **On the practical gain vs. GBDT:** We acknowledge that GBDTs are faster to train than neural networks, although in our experiments this difference is only about a factor of 6 (so not orders of magnitude, as one might think at first glance). As a result, running HPO for the same number of hyperparameter configuration trials is indeed faster for GBDTs. However, the ultimate mission is the accuracy a practitioner gets as the end product. In that sense GBDTs (e.g., XGBoost) are fast only at finding a sub-optimal prediction model, which is less appealing for a business-critical task, where a practitioner is willing to invest up to several CPU days on a single server (or less time with more compute power as our HPO is parallelizable), in order to achieve a statistically significant improvement for their application. In terms of the gain in accuracy wrt. XGBoost our proposed regularization cocktails actually also compare favorably for shorter runtimes, as shown by the new analytical results below:
>
>
> | Time (hours) | Wins | Ties | Losses | p-value |
> |:------------:|:----:|:----:|:------:|:-------:|
> |     0.25     |  16  |  3   |   15   |  0.964  |
> |     0.5      |  22  |  1   |   16   |  0.109  |
> |      1       |  23  |  1   |   15   |  0.029  |
> |      2       |  24  |  1   |   14   |  0.007  |
> |      4       |  25  |  2   |   13   |  0.003  |
> |      8       |  25  |  2   |   13   |  0.002  |
> |      16      |  25  |  2   |   13   |  0.001  |
> |      32      |  27  |  2   |   11   |  0.0009 |
> |      64      |  27  |  2   |   11   |  0.0009 |
> |      96      |  27  |  2   |   11   |  0.0008 |
>
>
> These results show the comparison of the performances of the best hyperparameter configurations that HPO found for the cocktails and XGBoost until the indicated time. The wins/losses/draws of our cocktails vs. XGBoost are aggregated over the collection of 40 datasets. We observe that given only 15 minutes the cocktails indeed tie XGBoost, and that they already **outperform XGBoost significantly after as little as one hour (on a single machine). Letting the HPO run for longer increases the gap further.**
>
>
> **On regularization:** Our definition of regularization abides to the common understanding that “regularization is the process of adding information in order to solve an ill-posed problem or to prevent overfitting” [a]. In that sense, weight decay, data augmentation, model selection, and other techniques that explicitly or implicitly fight overfitting and help generalization, are regularization measures ‘per definitionem’. We agree that we could extend the list of possible regularization techniques even further. However, we note that our current ingredients for the cocktail already suffice to yield better results than GBDT on the considered datasets.
>
> **On efficiency of our HPO:** There appears to be a misunderstanding on our HPO limit; this is 4 CPU days or 40xD=840 configurations, whichever is reached first. Indeed, for the smallest data set it only took as little as 1.38 hours to evaluate the 840 configurations. 4 CPU days is the maximum budget we allow for tuning hyper-parameters; however, this upper bound does not mean that the cocktails cannot find a reasonable hyper-parameter configuration for a smaller time budget. The analysis above shows that the cocktails outperform XGBoost significantly even after as little as 1 hour of HPO per dataset.
>
> **On AutoGluon:** AutoGluon Tabular is a very popular library for tabular datasets; however, we would like to draw attention to the fact that it is a fully-fledged AutoML system that combines i) various feature preprocessing techniques and ii) ensembles of {neural networks, random forests, nearest neighbors, GBDT, etc.,} in iii) a bagging and stacking style of ensembling. It is unfortunately a common misconception in the community that AutoGluon is a single method on its own, instead of it being an AutoML system consisting of diverse ensembles of other methods. In that sense, comparing the full AutoGluon system to GBDT means comparing a complex ensemble framework using many families of models to only one of these (i.e. GBDT). In contrast, our paper analyzes individual model families (i.e. neural networks vs GBDT) for tabular datasets, not the best way to ensemble model families (e.g. AutoGluon vs. other AutoML systems). Nevertheless, we agree that future work on building ensembles that include our regularization cocktails is a promising idea. For the sake of a principled comparison we limited AutoGluon to its neural network model family (official implementations of the library) and added it as a deep learning baseline. (We note that AutoGluon actually still has the unfair advantage of much more preprocessing and of stacking neural networks compared to the other baselines, such as TabNet, NODE, or our method). Nevertheless, our MLPs still significantly outperform the stacked+bagging MLPs of AutoGluon-Tabular.
>
> **On choice of regularizers:** We selected a pool of popular regularization methods of diverse types with no particular preference or discrimination, and stopped adding more when a critical mass developed. Please note that programming a pipeline that is able to apply 13 regularization methods simultaneously on Pytorch models was already a substantial software engineering effort. However, we do not see an intuition why adding even more regularizers in addition to the list of Table 1 might deteriorate results if sufficient time for HPO is available. (Only the anytime performance the reviewer, like many other people, cares about may deteriorate unless measures are taken to start the optimization from strong configurations.) Regarding your question on the importance of individual methods, the boring answer is “it depends on the dataset :)”. The HPO procedure filters out the unimportant methods if the respective “active=false” conditional hyperparameter is found to improve the validation accuracy on a specific dataset.
>
>
> [a] Bühlmann, Peter; Van De Geer, Sara (2011). "Statistics for High-Dimensional Data". Springer Series in Statistics: 9. doi:10.1007/978-3-642-20192-9. ISBN 978-3-642-20191-2.
>
> We hope that we could clarify your concerns and we would very much appreciate it if you considered increasing your score. Please feel free to raise any further point that was either left unanswered, or is a follow-up of our clarifications.

---

> > ### Comment · Reviewer_V9FE · 2021-08-30
> > **Thank you**
> >
> > Thank you for the considered rebuttal. The reviewers agree that these results are valuable to the community. I have upgraded my score.

---

### Decision · Program_Chairs · 2021-09-27

**Decision:**

Accept (Poster)

**Comment:**

The reviewers found this paper somewhat "unsurprising", but potentially high-impact and useful to the community. They also noted that papers like this one can be judged harshly, subjecting the authors to a potentially infinite number of experimental comparison. In this case, the consensus was that the revisions/responses from the authors did a good job addressing the concerns of the reviewers. We recommend adopting the suggestions arising from the discussion phase, particularly around the choice of title.